# Evaluation of Hydraulics and Downstream Fish Migration at Run-of-River Hydropower Plants with Horizontal Bar Rack Bypass Systems by Using CFD

Hannes Zöschg [1],*, Wolfgang Dobler [1], Markus Aufleger [1] and Bernhard Zeiringer [2]

1 Unit of Hydraulic Engineering, Department of Infrastructure, University of Innsbruck, 6020 Innsbruck, Austria; wolfgang.dobler@uibk.ac.at (W.D.); markus.aufleger@uibk.ac.at (M.A.)
2 Institute of Hydrobiology and Aquatic Ecosystem Management, University of Natural Resources and Life Sciences, 1180 Vienna, Austria; bernhard.zeiringer@boku.ac.at
* Correspondence: hannes.zoeschg@uibk.ac.at

**Abstract:** Anthropogenic structures often block or delay the downstream migration of fish in rivers, thereby affecting their populations. A potential solution at run-of-river hydropower plants (HPPs) is the construction of a fish guidance structure in combination with a bypass system located at its downstream end. Crucial to fish guidance efficiency and thus to fish behavior are the hydraulic flow conditions in front of the fish guidance structure and upstream of the bypass entrance, which have not thus far been investigated in depth. The present study aims to extend the knowledge about the flow conditions at these structures. Based on the results of 3D numerical simulations of two idealized block-type HPPs with horizontal bar rack bypass systems, the flow conditions were examined, and the fish guidance efficiency was predicted. Herein, a new method was used to represent the fish guidance structure in the numerical model. The results show that the approach flow to fish guidance structures at block-type HPPs varies significantly along their length, and areas with unfavorable flow conditions for downstream fish migration frequently occur according to common guidelines. Subsequently, eight variations were performed to investigate the effect of key components on the flow field, e.g., the bypass discharge. Finally, the results were compared with literature data and discussed.

**Keywords:** 3D numerical modelling; computational fluid dynamics; downstream fish migration; fish guidance structures; fish protection; flow field; horizontal bar rack bypass systems; hydraulics; hydropower plants

## 1. Introduction

In 2020, more electrical energy was generated in the EU by all renewables combined than by fossil fuels for the first time, thus taking an important step towards achieving the climate targets [1]. While the share of wind and solar energy is increasing rapidly, hydropower is still the largest renewable resource for electrical energy in the EU as well as worldwide [1]. However, the development of hydropower in Europe and particularly in the EU has been at a relatively low level since 2000, among other things due to the targets defined in the European Water Framework Directive 2000/60/EC (WFD) and even stricter national legislation [2,3]. Within the WFD, a "good" ecological status must be achieved for rivers in the EU [4]. Anthropogenic structures such as dams, weirs, or other barriers may affect the natural flow conditions in rivers [5–9], and hence block or delay the migration of fish [10–16]. In general, fish migrate in association with the use of resources that are not available in their current habitat, e.g., to reproduce, feed, or rest [17]. Therefore, fish migration in rivers is essential for the survival of diadromous species and some potamodromous species [5,18], and the construction of barriers can negatively influence the population or even lead to the extinction of entire species [19–21]. In this regard, the modernization of existing hydropower plants (HPPs) gains particular

importance, as it can be assumed that the modernization of an existing HPP can avoid most environmental impacts and conflicts compared to the construction of a new HPP on pristine and unregulated river stretches [3]. However, the average age of the EU's hydropower fleet was 42 years in 2019, taking into account HPPs that have already been retrofitted [2], which means that most HPPs were constructed at a time when the perception of fish migration was not the same as it is today. Consequently, retrofitting of old HPPs with adequate fish migration facilities can be very challenging, especially for downstream passage [22,23].

In order to enable fish migration past HPPs, different technical methods have been developed. In the past, measures were mainly limited to upstream fish migration, and downstream passage was mostly neglected [24]. In recent years, however, downstream passage has gained more attention [11,25–27]. Common methods used to facilitate downstream migration include (i) fish protection and bypass systems, (ii) fish-friendly turbines, (iii) fish-friendly operations, and (iv) fish collection systems [28]. In particular, the first represents the most frequent and, generally, the biologically most effective method [28]. Fish typically follow the main current, which usually leads to the turbines at HPPs [11,29]. By using fish guidance structures (FGSs) in combination with bypass systems located at the downstream end of the FGSs, turbine passages that often result in injury or mortality can be avoided and fish can be guided safely downstream of HPPs [11,28]. As FGSs, racks with small bar spacings slightly angled to the side or inclined to the riverbed (<45°) can be used [11,28,30]. Since fish that prefer to migrate near the riverbed (e.g., eels [31,32]) are forced to leave their preferred flow depth at racks inclined to the riverbed and have to enter the bypass near the water surface, racks angled to the side are particularly recommended [33]. For example, in Sweden [25] and Switzerland [34], angled racks with bypasses are considered best-practice solutions for downstream passage. Furthermore, previous experimental ethohydraulic tests indicated that racks angled to the side have a more favorable guiding effect compared to racks inclined to the riverbed [31,35,36]. The bars of angled racks can either be horizontally oriented (so-called horizontal bar racks, short HBRs) or vertically oriented (vertical bar racks, VBRs). If a bypass is located at the downstream end of an HBR to allow for safe downstream passage, these systems are referred as horizontal bar rack bypass systems (HBR-BSs). HBR-BSs have already been installed at more than 100 small- to medium-sized HPPs with design discharges $Q_d < 120$ m$^3$/s in Europe [37]. During operation, HBR-BSs can be beneficial due to small head losses [37,38] and the potential to pass floating debris automatically downstream through the bypass [28,37]. State-of-the-art reviews of HBRs and HBR-BSs can be found in Meister [37] and Maddahi et al. [39].

For the design of adequate HBR-BSs, as well as other FGSs with bypass systems, favorable approach flow conditions are essential, including (i) the angle of the approach flow vector, (ii) flow velocities, (iii) velocity gradients, and (iv) turbulent flow structures [28,40]. Current design guidelines were developed on the basis of practical experiences at pilot HPPs [41] and focus primarily on the former two flow parameters. To guide fish along the rack to the bypass, the rack parallel velocity component $v_p$ should be larger than the rack normal velocity component $v_n$ ($v_p/v_n > 1$) [11,28,42]. Therefore, HBRs are usually designed angled to the side. For example, Ebel [28] recommends rack angles between $\alpha = 20°$ and $40°$ to the unaffected flow direction for high fish guidance efficiencies. To avoid fish being impinged against the rack, $v_n$ should be lower or equal than the sustainable swimming speed $v_{sus}$ ($v_n \leq v_{sus}$), where $v_{sus}$ is the aerobic swimming activity that fish can sustain for more than 200 min without fatigue, depending on fish length and water temperature [26,28]. Moreover, the flow velocities at the bypass entrance also play an important role, since an efficient bypass is considered to be a key point for the successful design of downstream migration facilities [30,39,43–45]. The flow conditions at the bypass entrance are often described in design guidelines with the relative bypass discharge $Q_{by,rel}$ as the ratio of the bypass discharge $Q_{by}$ to the design discharge $Q_d$ ($Q_{by,rel} = Q_{by}/Q_d$). In order to obtain moderate flow velocities at the bypass entrance and thus favorable flow accelerations for fish downstream passage, a minimum $Q_{by,rel}$ between 2% and 5% is recommended for HBR-BSs [28]. By following this recommendation, the ratio between the velocity at the bypass entrance $v_{by}$ and the mean approach flow velocity $v_0$

should be between $v_{by}/v_0$ = 1.0 and 1.5, in trout waters even up to 2.0, which is considered to be an attractive value to guide fish into the bypass [34]. Similar values in this range are recommended by other authors, e.g., Meister [37], who obtained higher fish guidance efficiencies at $v_{by}/v_0$ = 1.2 in ethohydraulic experiments with HBR-BSs.

The design of FGSs and bypass systems should be based on the target species and stage-specific aspects of fish that have to be protected [26,28,30,43]. However, there are still knowledge gaps about fish behavior as well as the required hydraulic and geometric parameters, especially relating to potamodromous species [29,30,46]. Fish swimming behavior is, among other stimuli (e.g., visual, acoustic), a behavioral response to hydraulic conditions [21]. It is known that fish avoid areas with abrupt changes in velocity gradients, both acceleration and deceleration, as well as strong turbulent structures [12,24,29,47–50]. Common guidelines often cannot predict the complexity of spatial flow conditions in front of FGSs and at the bypass entrance. As a result, FGSs with bypass systems planned according to common guidelines are rarely 100% effective [26]. Instead, the "trial and error" approach is often used [24], occasionally followed by revisions, which can be very time-consuming and expensive [51]. Subsequently, the still lacking knowledge about efficient downstream migration facilities is a challenge for HPP designers and operators as well as for authorities [30,52].

Computational fluid dynamics (CFD) can be used to determine flow conditions where measurements (e.g., with acoustic Doppler current profilers) are limited due to safety or operational constraints [53]. While CFD has been applied multiple times to study and optimize flow conditions at fish upstream migration facilities (e.g., [53–57]), it has been used for downstream migration facilities only in a few cases [27], and if so, these studies were mostly limited to specific sites. In recent years, several studies have been conducted to investigate typical fish migration patterns and route choices at HPPs using telemetry data combined with the results of 3D numerical simulations to identify connections between fish behavior and flow conditions (e.g., [21,47,58,59]). Such knowledge can be used for the development of more efficient migration facilities [58]. Furthermore, a few authors implemented CFD to investigate measures to optimize fish guidance efficiency for downstream migration at HPPs [27,51,60–64]. In particular, Feigenwinter et al. [27] developed a conceptual approach for positioning FGSs at HPPs based on the results of 3D numerical simulations at cross sections of potential FGS locations in combination with fish biology and expert knowledge. Similar to most common guidelines, the criteria used for the evaluation were predominantly $v_p/v_n > 1$ and $v_n \leq v_{sus}$, while other flow parameters such as abrupt velocity changes and turbulent flow structures were not taken into account. In addition, since the bypass was excluded in the approach, the flow conditions at the bypass entrance could not be investigated.

The present study aims to extend the knowledge about the hydraulic conditions and the related fish behavior at HBR-BSs at run-of-river HPPs. In detail, two idealized HPPs were designed exemplarily for the Austrian catchment area of the Danube River, which are based on the design of existing HPPs in the block-type layout without appropriate measures for downstream fish migration, and equipped with HBR-BSs. After 3D numerical simulation, substantial flow parameters including velocity gradients and turbulent flow structures were evaluated and the fish guidance efficiency (FGE) assessed. Herein, the focus of the study was on the area upstream of the HBR and at the bypass entrance. Furthermore, geometric variations of key components were performed to examine their effect on hydraulics and downstream fish migration. As a result, not only was the knowledge for the design of efficient downstream migration facilities enhanced, but the presented approach can also be used in the design process of new HPPs as well as for the modernization of existing HPPs without appropriate measures for downstream fish migration, both in the rivers investigated in this study and in other rivers after adapting site- and fish-specific parameters.

## 2. Materials and Methods

### 2.1. Studied HPPs

#### 2.1.1. Initial Designs

HPPs differ significantly due to site-specific conditions and are unique in terms of design [1]. In order to examine a wide range of both existing and newly constructed run-of-river HPPs, idealized HPPs with typical design components based on existing ones in Central Europe were used in this study. Generally, run-of-river HPPs have a weir for damming the river and a powerhouse for electricity production. For rivers where shipping plays a role, a navigation lock is often provided in the vicinity of the weir [65]. In addition, most HPPs these days are equipped with an appropriate fishway to create a migration corridor for upstream fish migration, e.g., a near-natural fish pass. A typical design of such an HPP in the block-type layout with the components powerhouse, weir, navigation lock, and fishway is shown in Figure 1a, while Figure 1b shows the same HPP after modernization with appropriate measures for downstream fish migration.

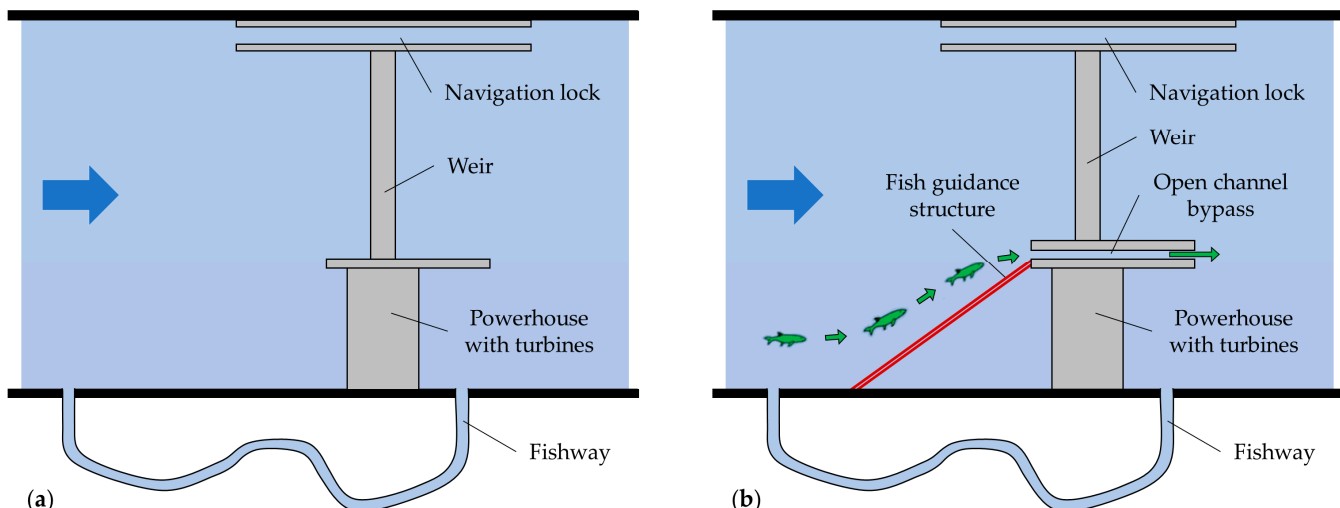

**(a)**          **(b)**

**Figure 1.** Typical schematic layout of a block-type hydropower plant in Central Europe (**a**) without and (**b**) with measures for fish protection.

For the design approach of the HPPs in this study, (i) potential sites as well as the layout of the idealized HPPs were defined, (ii) the HPPs were equipped with measures for downstream fish migration according to common guidelines, and (iii) 3D numerical simulations of the HPPs were conducted, and, based on the results, geometric variations were performed.

In the first step, two potential sites in the Austrian catchment area of the Danube River were defined to examine both a small and a medium-sized river. The first is located at a pre-alpine river with an (assumed) total river discharge $Q_0 = 50$ m$^3$/s, typically in the grayling region (Hyporhithral), and the second at an alpine river with $Q_0 = 10$ m$^3$/s, typically in the lower trout region (Metarhithral). In this regard, a greater diversity of fish species generally occurs at the site of the pre-alpine river than at that of the alpine river [66]. Subsequently, block-type HPPs were projected for both sites, with the powerhouse located next to the weir. Herein, only the weir and the powerhouse were considered among the components shown in Figure 1a, since it can be assumed that both the navigation lock and fishway have no or only a negligible effect on the hydraulic conditions in the area of interest for this study. The powerhouse and weir are separated by a dividing pier, which was designed according to the design recommendations of Häusler [67] to optimize the flow conditions towards the turbines for the HPP Landau on the Isar River in Germany. In front of the turbine inlets, the concrete bottom is inclined downward. In addition, more simplifications were made in the design process of the associated 3D models, used for the numerical simulations, to reduce the complexity of the HPPs, and thus also the computational costs. For instance, the

section downstream of the weir and turbines was not modeled. Moreover, the banks at all sites are designed simplified vertical, and the soil has no slope. The main characteristics and parameters of these HPPs are listed in Table 1.

**Table 1.** Characteristics and main parameters of the studied hydropower plants (HPPs). Varied values of parameters that have been modified within the variations are marked with *.

| Site | Pre-Alpine River | Alpine River |
|---|---|---|
| Fish Zonation | Grayling Region | Lower Trout Region |
| HPP Construction Type | Block-Type | Block-Type |
| Total river discharge $Q_0$ [m$^3$/s] | 50 | 10 |
| Design discharge $Q_d$ [m$^3$/s] | 48 | 8 *, 9 *, 9.5 |
| Bypass discharge $Q_{by}$ [m$^3$/s] | 2 | 0.5, 1 *, 2 * |
| Mean approach flow velocity $v_0$ [m/s] | 0.36 | 0.25 |
| Mean rack normal velocity component $\overline{v_n}$ [m/s] | 0.46 | 0.24 *, 0.45 |
| River width $w_0$ [m] | 35 | 20 |
| Bypass width $w_{by}$ [m] | 1 | 0.5 |
| Length of the FGS $l_{FGS}$ [m] | 23.34 *, 25.94 | 10.58, 20.19 * |
| Approach water level upstream of the HPP $h_0$ [m] | 4 | 2 |
| Rack angle $\alpha$ [°] | 40 | 20 *, 40 |

After the conceptual design of the "existing" HPP components, supposed suitable FGSs, in combination with bypass systems, were implemented. HBR-BSs were chosen for this application. The principle design used for the HBR-BSs is shown in Figure 2 and is based on the angled bar rack bypass system of Ebel, Gluch, and Kehl [28].

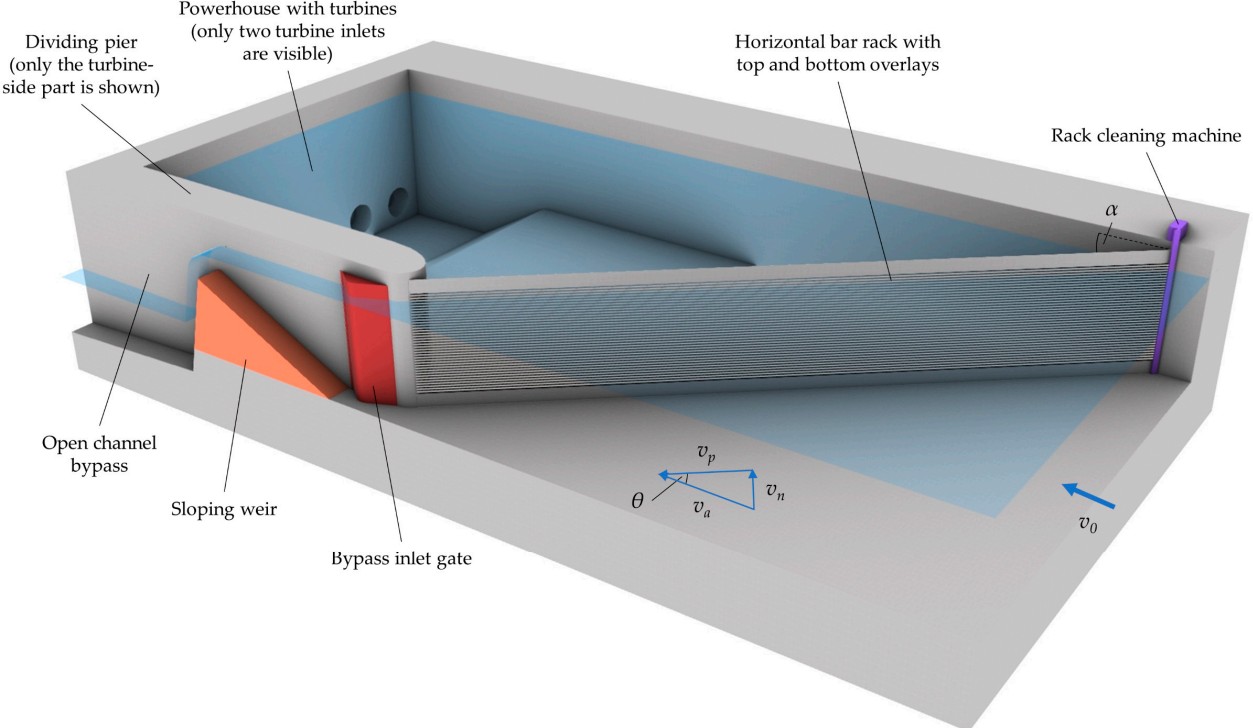

**Figure 2.** Schematic representation of the horizontal bar rack bypass system (HBR-BS) concept used in this study, adapted from Ebel [28] and Maddahi et al. [39], with $\alpha$ = horizontal rack angle, $\theta$ = horizontal angle between the approach flow and the rack, $v_a$ = approach flow velocity to the rack, $v_n$ = rack normal velocity component, $v_p$ = rack parallel velocity component, and $v_0$ = mean approach flow velocity.

In terms of a simplified assumption, HBRs with circular bar shape and a relatively high blocking ratio of 50% were chosen for all designs in this study to cause a large impact on the flow conditions due to the FGSs. Generally, the geometric representation of FGSs in numerical simulations requires a fine mesh resolution, especially for fully accounting for all effects of the bars on the flow conditions, which leads to very high computational costs [27]. Therefore, a simplified method for the representation of the FGS was used in this study, described in Section 2.2.4. In addition to this, the rack cleaning machine as well as top and bottom overlays were neglected in the designs, except for variation V7, in which the latter was considered (Section 2.1.2).

The open channel bypass was located at the downstream end of the FGS to provide a descent corridor over the entire water depth for fish migrating along the FGS. Open channel bypasses are recommended over pressurized pipe bypasses for practical reasons, especially clogging [28,37]. The bypass system consisted of an inlet gate followed by a sloping weir. The inlet gate is intended to create a favorable attraction flow with moderate velocities. In addition, it should prevent fish from leaving the bypass in the upstream direction after they have entered the bypass, and it can be opened for rack cleaning or flushing [28,39]. At the HPPs in this study, the inlet gate was open across the entire water column and thus, contrary to near-bottom and near-surface openings, offers the advantage that fish do not have to leave their specific swimming depth [68]. The inlet gate was fixed, measured half the width of the bypass (i.e., 0.5 m opening width at 1.0 m bypass width), and was shaped for favorable flow conditions as recommended by guidelines [34] (Figure 2). Note that the vertical axis inlet gate shown in Figure 2 was installed on the turbine-side part of the dividing pier due to illustration issues, while the inlet gate in the numerical models of this study is located on the weir-side part, except for variation V3 (Section 2.1.2). Generally, the sloping weir controls the discharge in the bypass and can be fixed or adjusted to varying discharge conditions. Furthermore, it also prevents fish from returning in the upstream direction after successful downstream migration [28]. In this study, the sloping weir was assumed to be fixed, and its design was created according to common guidelines, e.g., the slope of the weir was within the recommended range of 10 to 30° [28].

### 2.1.2. Variations

Following the numerical simulations of the HPPs described in Section 2.1.1, geometric variations (V1–V8) were performed, which are listed in Table 2. It should be noted that the variations provide only a sample of the very large number of possible variations and were conducted for only one of the two HPPs in this study. They were selected based on the results of the initial designs to examine some fundamental questions regarding the influence of geometric changes on the flow field and further to assess the associated behavioral correlations.

### 2.1.3. Examined Operating Case

All studied HPPs were considered the most demanding situation for downstream migrating fish, in which the weir is closed and the turbines are in full operation [27]. Consequently, fish are only able to migrate downstream through the turbines or the bypass, and the angle of the approach flow vectors and flow velocities to the FGS, as well as the velocity gradients and turbulent flow structures at the bypass entrance, are assumed to be the worst case. $Q_0$ is divided into a small part for $Q_{by}$ and a large part for $Q_d$ ($Q_0 = Q_d + Q_{by}$). The corresponding values are listed in Table 1. Note that the share of $Q_0$ flowing through the fishway is neglected here, since the fishway was not included in the numerical models.

**Table 2.** Studied variations V1–V8 with description of varied components.

| Variation | Varied Component | Description | Schematic Illustration | HPP |
|---|---|---|---|---|
| V1 | Dividing pier (weir-side part) | Shifted 1.0 m in the upstream direction |  | Pre-alpine river |
| V2 | Dividing pier (weir-side part) | Modified the shape and width for smoother flow conditions around the pier |  | Pre-alpine river |
| V3 | Inlet gate | Installed at the turbine-side part of the dividing pier |  | Pre-alpine river |
| V4 | Sloping weir | Lowered to double the bypass discharge $Q_{by}$ compared to the initial design |  | Alpine river |
| V5 | Sloping weir | Lowered to quadruple the bypass discharge $Q_{by}$ compared to the initial design |  | Alpine river |
| V6 | Fish guidance structure | Modified the rack angle $\alpha$ to 20° |  | Alpine river |
| V7 | Fish guidance structure | Implementation of a bottom overlay with a height of 0.2 m |  | Alpine river |
| V8 | Fish guidance structure | Integrated into the headrace channel with the bypass on the orographic right side (cf. Figure 2) |  | Pre-alpine river |

### 2.2. Numerical Models

#### 2.2.1. General

For this study, the software ANSYS Fluent 19, based on the finite volume approach, was used to simulate the 3D hydraulic conditions. The Semi-Implicit Method for Pressure-Linked Equations (SIMPLE) algorithm solves the Reynolds-averaged Navier-Stokes (RANS)

equations, which are considered a convenient trade-off between accuracy and computational cost [56]. The turbulent flow is described by using the realizable $k$–$\varepsilon$ turbulence model with scalable wall functions. The RANS method in combination with the $k$-$\varepsilon$ turbulence model has been implemented in similar previous studies (e.g., [27,51,62]). The free surface represented by the interface between air and water was modelled using the volume of fluid (VOF) method [69] to account for water surface fluctuations that may affect the flow field, with the water fraction in each element expressed by the water volume fraction parameter $\alpha_w$. Further details regarding the fundamental governing equations can be found in the ANSYS Fluent Theory Guide [70].

### 2.2.2. Boundary Conditions

At the inlet of each model domain, a constant inflow velocity ($v_0$, Table 1) was defined as the boundary condition. Water flows out of the domains through the turbines and the bypass. Constant values for the outflow discharges ($Q_{by}$ and $Q_d$, Table 1) were used at both locations. In order to achieve more homogeneous outflow conditions in the cross-section of the turbines, the turbines were simplified using circular cylinders, each extended by one meter in the downstream direction and having the identical constant outflow discharge at the end. The downstream domain end at the bypass was located a few meters downstream of the sloping weir (Figure 2) to include effects due to water flowing over the weir. A symmetry boundary condition was applied at the top of the air-domain. All other boundaries were set to no-slip walls.

### 2.2.3. Spatial and Temporal Discretization

Unstructured meshes were used for the numerical simulations, consisting of tetrahedra, except for the simplified representation of the FGSs, which consisted of hexahedra (Section 2.2.4). The domains were divided into two regions with different mesh resolutions. Since the focus of the study was on the area in front of the FGS and at the bypass entrance, a maximum element face sizing (MFS) of 0.2 m was used in this region, while the MFS in the outer region was 0.4 m. The minimum face sizing was defined as 0.01 m in all regions. Inflation layers with a first layer thickness of 0.005 m were used to account for the effects of wall roughness on the flow field. The chosen mesh resolution was defined based on a mesh independency study according to the American Society of Mechanical Engineers (ASME) criteria [71], described in Appendix A. In total, the number of elements in the meshes ranged from 800,000 to 2,700,000 elements depending on the HPP or variation studied.

During the numerical simulations, the time step was gradually increased from 0.0001 s to 0.02 s to achieve a well-balanced compromise between adequate computation time and robust computation. For the definition of the maximum time step, several simulations were performed in which the time step was varied, and the results were compared to confirm their independency from the chosen maximum time step. In total, the simulations were performed for 900 s to ensure that the simulations converged to a stable solution. To verify this, the mass balance of water and the mean flow velocities in L1–L5 were monitored over time. A steady state solution was assumed when nearly constant values were reached for these hydraulic parameters.

### 2.2.4. Implementation of the Fish Guidance Structure in the Numerical Model

For the representation of the FGS in the numerical model, a porous medium with an integrated function to account for the angle-dependent approach flow conditions was used as a simplified method. During the construction of the domain, a body with a thickness of 0.05 m was created in the plane of the FGS. After the spatial discretization, this body consisted of hexahedral elements with dimensions of 0.20 m $\times$ 0.20 m $\times$ 0.05 m. A user-defined function (UDF) was developed to consider the different approach flow conditions to FGSs. During the numerical simulations, the UDF continuously calculated the horizontal angle between the approach flow and the rack $\theta$ for each element in the body and calculated the pressure drop $\Delta p$ in the element. This value is based on head loss

coefficients $\xi$ depending on the rack configuration studied, which were estimated using the modified formula of Meusburger [72] provided by Böttcher et al. [73]. The schematic workflow of the porous medium and the developed UDF is shown in Figure 3.

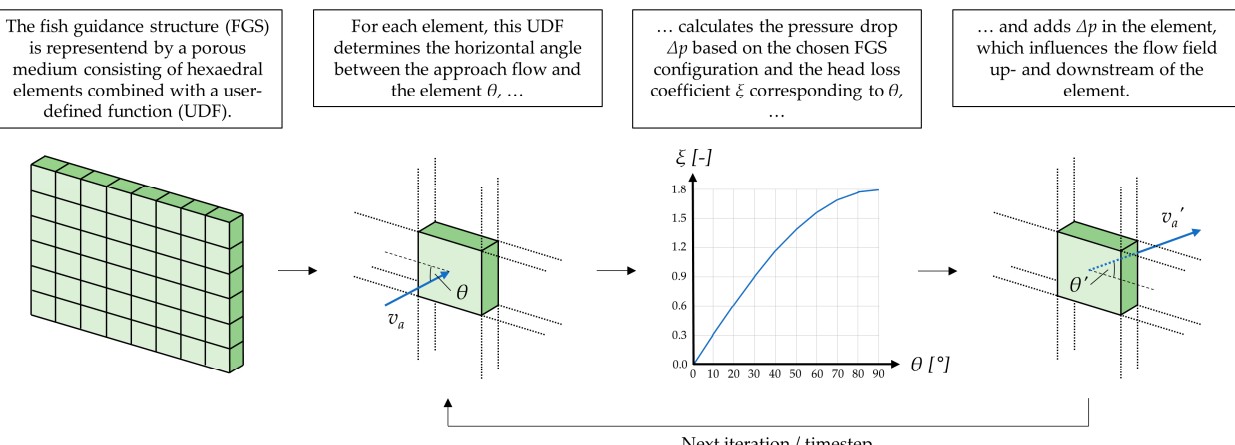

**Figure 3.** Schematic workflow of the porous medium with the developed UDF during a numerical simulation, with $v_a$ = approach flow velocity to the FGS, $v_a'$ = outflow velocity downstream of the FGS, and $\theta'$ = horizontal angle between the outflow and FGS downstream of the FGS.

By inserting a specific angle-dependent value for $\Delta p$ in each element of the body, the effects HBRs have on the flow field can be considered in a simplified way, including the slightly horizontal flow deflections observed in previous model tests [41]. To verify this method, the difference of $\Delta p$ upstream and downstream of the porous medium were determined based on the results of the 3D numerical simulations and compared to the calculated $\Delta p$.

### 2.3. Criteria Used for the Evaluation

Based on the results of the 3D numerical simulations, the potential fish behavior was estimated, and the FGE was evaluated. For this, it was assumed that the fish behavior during downstream migration depends exclusively on the hydraulic conditions. Other factors to which fish may react sensitively or the motivation of fish to migrate downstream were not considered.

In guidelines, a favorable FGE along the FGS is assumed to exist if the criterion $v_p/v_n > 1$ is fulfilled [11,28,42]. In this case, the horizontal angle between the approach flow and the FGS $\theta$ is smaller than 45° (Figure 2). Consequently, the main flow direction occurs along the FGS, and fish are guided towards its downstream end and further into the bypass. Moreover, impingement to the FGS or entering the headrace channel can be avoided as long as acceptable values of $v_n$ occur, which should be less than or equal to $v_{sus}$ ($v_n \leq v_{sus}$) [26,28]. Turnpenny and O'Keeffe [26] recommended that 90% of downstream migrating fish should be able to swim against $v_n$ for at least 200 min without being impinged against the FGS. Therefore, the critical swimming speed in many guidelines is often defined with 200 min. Besides $v_{sus}$, the swimming speed of fish can also be described by the prolonged swimming speed $v_{pro}$, which can be maintained for 1 to 200 min without exhaustion [74]. Hereafter, the swimming duration $t$ of $v_{pro}$ is defined as 1 min. To determine fish swimming speed $v_f$, Ebel [28] proposed multivariate models for European fish species based on literature data, distinguishing between a general model and specific models for rheophilic and non-rheophilic species. For the study area in the Austrian catchment area of the Danube river, the model for rheophilic species is of particular relevance, defined as

$$\log(v_f) = 0.5460 + 0.7937 \log(TL) - 0.0902 \log(t) + 0.2813 \, \log(T), \tag{1}$$

where *TL* is the total length of the fish and *T* the water temperature. This model can be used for most rheophilic fish species whose swimming style correspond interspecifically. For other species with differing swimming styles such as lampreys, eels, and sturgeons, species-specific models have been developed [28]. However, these species are not relevant to the present study area. Figure 4 shows $v_{pro}$ and $v_{sus}$, calculated using Equation (1), as a function of *TL*. For the definition of *T*, typical values for mean summer water temperatures (July to September) in Austrian rivers in the grayling region (Hyporhithral) and lower trout region (Metarhithral) with 8 to 14 °C and 5 to 10 °C, respectively, were used [75]. Based on this, *T* at the HPP on the pre-alpine river was defined as 11 °C, and at the HPP on the alpine river as 8 °C.

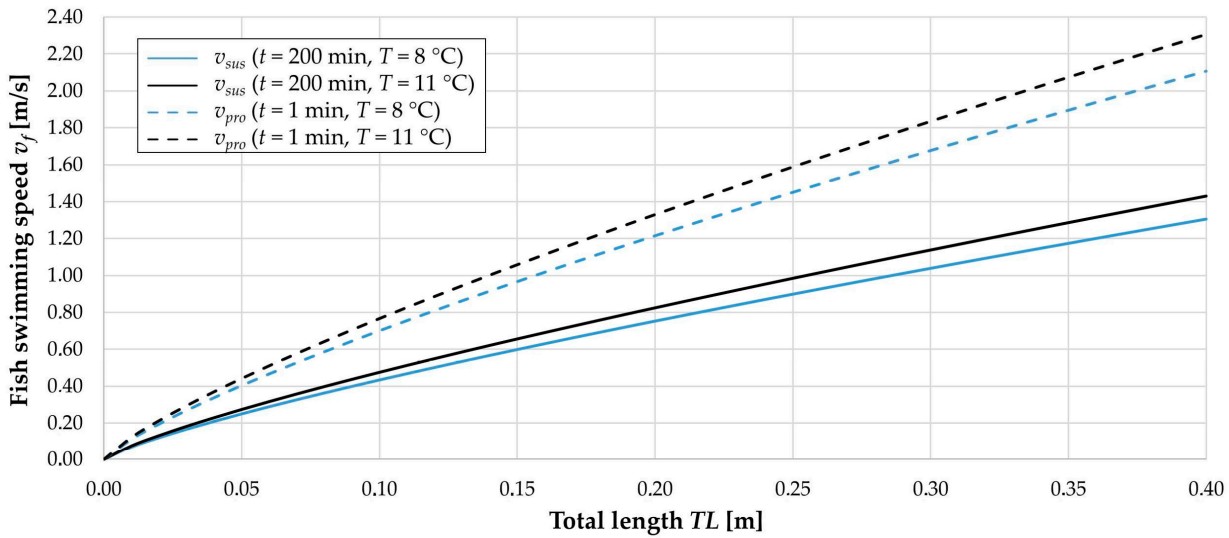

**Figure 4.** Sustained swimming speed $v_{sus}$ and prolonged swimming speed $v_{pro}$ depending on the total length TL, calculated with Equation (1), with *t* = swimming duration, and *T* = water temperature.

At the downstream end of the FGS, an attraction flow into the bypass should be provided, i.e., a uniform slight increase in flow velocity in the direction of the bypass [28,34,37,43]. However, if the velocity is increased too rapidly, fish may exhibit an avoidance response [12,40,48,50], which also applies for abrupt deceleration [76,77]. Therefore, complex flow conditions with spatially and temporally rapidly changing velocities [48] need to be avoided. In this regard, turbulent kinetic energy *TKE* and spatial velocity gradients *SVG* were considered as part of the evaluation. High values for *TKE* can cause disorientation in fish and reduce swimming efficiency [47,48,78,79]. Generally, *TKE* is calculated as

$$TKE = \frac{1}{2}\left(\overline{u'^2} + \overline{v'^2} + \overline{w'^2}\right), \tag{2}$$

where $u'$, $v'$, and $w'$ are the velocity fluctuations in the *x*-, *y*-,and *z*-directions, respectively. Furthermore, by avoiding areas with high values of *SVG*, fish can minimize predation risk, physical injuries due to increasing velocities, and migration delays due to decreasing velocities [76]. *SVG* is defined as

$$SVG = \begin{pmatrix} \dfrac{\partial u}{\partial x} & \dfrac{\partial u}{\partial y} & \dfrac{\partial u}{\partial z} \\ \dfrac{\partial v}{\partial x} & \dfrac{\partial v}{\partial y} & \dfrac{\partial v}{\partial z} \\ \dfrac{\partial w}{\partial x} & \dfrac{\partial w}{\partial y} & \dfrac{\partial w}{\partial z} \end{pmatrix}, \tag{3}$$

where *u*, *v*, and *w* are the local flow velocities in the *x*-, *y*-, and *z*-directions, respectively. Using Equation (3), *SVG* can be visualized straightforwardly as the output of the numerical

simulations. In contrast, in previous ethohydraulic laboratory studies [12,50,76], *SVG* was determined as the velocity gradient related to *TL* and the position of the fish at the time of the avoidance response, using

$$SVG_f = \frac{|v_H - v_T|}{TL}, \tag{4}$$

where $SVG_f$ is the spatial velocity gradient experienced by fish (and expressed in cm/s/cm or 1/s), $v_H$ is the velocity at the head of the fish, and $v_T$ is the velocity at its tail. Note that different notations were used for *SVG* to distinguish between Equation (3) (used in general) and Equation (4) (applied to fish). Since fish are known to orient themselves in the flow in a streamwise direction to conserve energy [10], it was assumed that a fish would align itself in the direction of a flow vector, but not perpendicular to it. Based on this, the normalized velocity vectors from the results of the numerical simulations were used to estimate $SVG_f$. Herein, the velocities at the tip of the considered vector and at its tail were used for $v_H$ and $v_T$, respectively, and its length for *TL*. As Equation (4) uses the absolute value of the difference between $v_H$ and $v_T$, the velocity gradient can be determined regardless of the orientation of the fish in either positive (tail first) or negative (head first) rheotaxis.

In general, not much is known about fish behavior affected by complex flow conditions [12,31,34,48]. Therefore, no critical values were defined for the evaluation of *TKE*, *SVG*, and $SVG_f$ in this study. Rather, only enhanced values were pointed out, and relative considerations between initial designs and variations were made. However, the values obtained will be compared with literature data in Section 4.1.

To perform the evaluation as consistently as possible for all HPPs, three horizontal planes were defined for the analysis of the hydraulic parameters described above, which are (i) near the riverbed (0.1 m above the riverbed), (ii) mid-flow depth ($z/h_0 = 0.5$), and (iii) near the water surface (0.5 m below the water surface). Thus, the swimming depths of different fish species could be considered individually. Further, to evaluate the flow conditions upstream of the FGS, a vertical plane 0.1 m in front of the FGS was used, considering only areas where $\alpha_w$ was equal or higher than 0.5.

## 3. Results

### 3.1. Overview

Since the results of the two initial HPP designs show similar patterns, only the results of the HPP on the pre-alpine river are described in detail in Section 3.2. Subsequently, the principal results of variations V1–V8 are presented in Section 3.3, including basic findings of the HPP on the alpine river for comparison where needed. Further information on the geometric and hydraulic preconditions, as well as additional results of the numerical simulations of all studied HPPs, are provided as Supplementary Materials.

### 3.2. HPP on the Pre-Alpine River

#### 3.2.1. Overall Flow Field

Figure 5 shows the velocity field around the FGS of the HPP on the pre-alpine river at mid-flow depth ($z = 2.0$ m, $z/h_0 = 0.5$). Note that the green line indicates the position of the FGS in the model. In general, similar flow patterns occur near the riverbed at $z = 0.1$ m ($z/h_0 = 0.025$), in mid-flow depth at $z = 2.0$ m ($z/h_0 = 0.5$), and near the water surface at $z = 3.5$ m ($z/h_0 = 0.875$). However, the most relevant single deviation for this study appears at the bypass entrance, described in Section 3.2.3. As expected for a block-type HPP, the flow in the area upstream of the HPP is deflected towards the headrace channel, and the velocities increase due to the reduced width of the channel compared to the overall width of the river upstream of the HPP (Figure 5a). Thus, the velocities towards the FGS also increase continuously. The flow gets slightly deflected by the FGS, in front of the FGS slightly parallel to the FGS, which can be observed particularly at the downstream end of the FGS, and behind the FGS slightly in the direction perpendicular to the FGS. The free water surface at $\alpha_w = 0.5$ increases in front of the FGS and decreases after the FGS depending on the local velocity distribution. Therefore, the largest difference between the

water level upstream and downstream of the FGS ($\Delta h = 0.05$ m) occurs in the area of highest local velocities close to the downstream end of the FGS. At lower velocities, the effect of the FGS on the water level is negligible. Downstream of the FGS, the flow velocities increase continuously on the orographic right side of the headrace channel. On the orographic left side, the highest velocity in the numerical model ($v_{m,max} = 1.38$ m/s), excluding the increased flow velocities in the bypass as an effect of the sloping weir, occurs next to the upstream end of the turbine-side part of the dividing pier at $z = 3.5$ m ($z/h_0 = 0.875$). Further downstream, the flow becomes more homogenous over the width of the headrace channel. However, the headrace channel was designed too short to allow uniform flow conditions to establish themselves towards the turbines. Compared to the flow deflection caused by the block-type layout, the FGS has a minor effect on the turbine approach flow. Moreover, in the vicinity of the weir, flow-calmed areas with relatively low flow velocities occur.

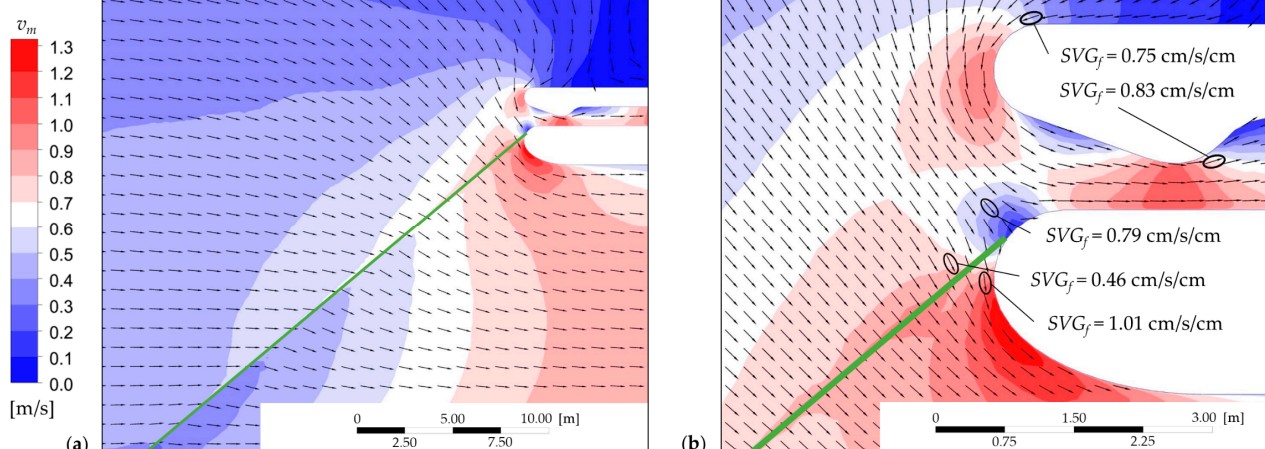

**Figure 5.** Velocity field at the HPP on the pre-alpine river at $z = 2.0$ m ($z/h_0 = 0.5$): (**a**) velocity magnitude $v_m$ and normalized flow vectors with a length of 0.75 m in a rectangular grid of 1 m distance around the FGS, and (**b**) velocity magnitude $v_m$, normalized flow vectors with a length of 0.2 m in a rectangular grid of 0.2 m distance and selected values for the spatial velocity gradient experienced by a fish $SVG_f$ with total length $TL = 0.2$ m at the bypass entrance. The green line indicates the position of the FGS in the model.

### 3.2.2. Hydraulic Parameters in Front of the FGS

For the evaluation of FGE along the FGS, the hydraulic parameters 0.1 m in front of the FGS were examined (Figure 6). Generally, at small distances to the FGS (up to 0.5 m), no significant differences in flow patterns were observed in the results of the numerical simulations. In Figure 6, $x_{FGS}$ is defined as the distance from the downstream end ($x_{FGS} = 0$ m) along the FGS to the upstream end ($x_{FGS} = 25.94$ m), and $z$ is the distance from the riverbed to the water surface (at $\alpha_w = 0.5$). The velocities $v_a$ (Figure 6a) and $v_n$ (Figure 6b) increase gradually along the FGS in the direction of the bypass, with local maxima close to the downstream end, where $v_{a,max} = 1.03$ m/s at $z = 0.06$ m ($z/h_0 = 0.03$) and $v_{n,max} = 0.87$ m/s at $z = 2.0$ m ($z/h_0 = 0.5$), respectively. Both values are above the calculated $v_{pro} = 0.83$ m/s for fish with $TL = 0.11$ m (Figure 4). Since the velocities converge to zero at the walls, minima of the flow velocities can only be estimated ($v_{a,min} \approx 0.3$ m/s and $v_{n,min} \approx 0.2$ m/s, respectively, close to the right bank). In comparison, the mean $v_n$, calculated as the ratio of $Q_d = 48$ m³/s and the hydraulically active area of the FGS $A_{FGS,hyd} = 103.76$ m², is $\overline{v_n} = 0.46$ m/s. Figure 6c shows $v_p$ along the FGS. Note that $v_p$ is defined positive in the direction of the bypass, while negative values imply a rack parallel velocity component along the FGS towards its upstream end. This makes it possible to distinguish between favorable flow conditions towards the bypass ($v_p > 0$) and unfavorable ones in the opposite direction ($v_p < 0$). At $x_{FGS} \approx 18$ to 20 m, $v_{p,max} = 0.44$ m/s occurs. From there towards the downstream end of the FGS, $v_p$ decreases, with negative values

occurring at the downstream end, especially near the bottom. This leads to the absence of a guiding effect in the direction of the bypass over the entire water depth in this area, which can also be observed by means of the velocity vectors in Figure 5b. Consequently, fish must actively swim against the rack parallel flow to reach the bypass as they migrate along the FGS. The ratio $v_p/v_n$ has favorable values above 1 only near the upstream end of the FGS (Figure 6d). Thus, this indicates that without further optimization, favorable conditions for downstream migration of fish along the FGS will not occur. Nevertheless, the FGS leads to relatively small increases in *TKE* and *SVG* 0.1 m in front of the FGS, as shown exemplarily in Figure 6e for *TKE*. Although the smaller hydraulically active area of the FGS $A_{FGS,hyd}$ results in locally increased velocities and thus also increased *SVG*, the latter effects are only very local, as can be seen in Figure 7b, where *SVG* was calculated with Equation (3). Therefore, the hydraulic parameter *SVG* 0.1 m in front of the FGS is not included in Figure 6.

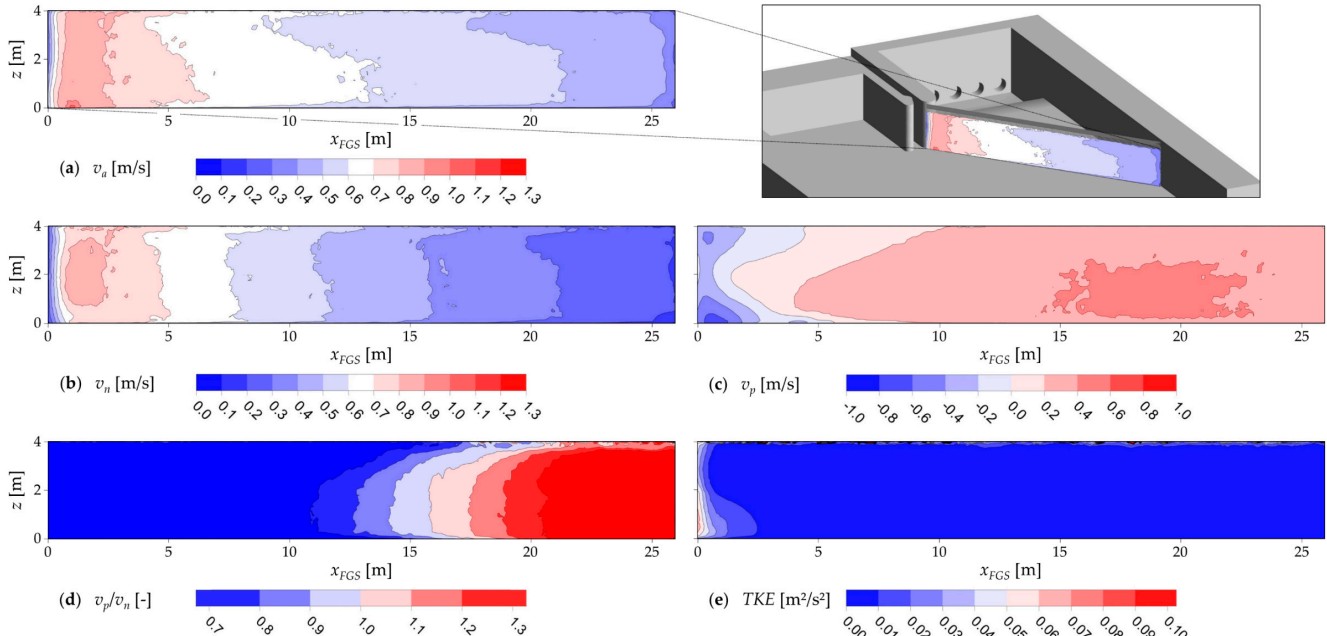

**Figure 6.** Hydraulic parameters at the HPP on the pre-alpine river 0.1 m in front of the FGS: (**a**) approach flow velocity to the FGS $v_a$, (**b**) rack normal velocity component $v_n$, (**c**) rack parallel velocity component $v_p$ (positive in the bypass direction, negative in the direction of the upstream end of the FGS), (**d**) ratio of $v_p/v_n$, and (**e**) turbulent kinetic energy *TKE*.

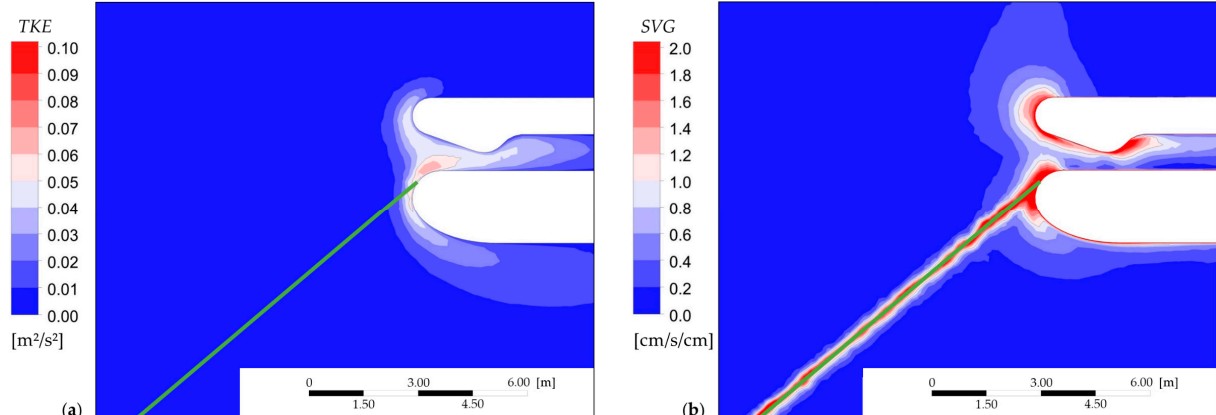

**Figure 7.** Hydraulic parameters at the downstream end of the FGS and the bypass entrance of the HPP on the pre-alpine river at $z = 2.0$ m ($z/h_0 = 0.5$): (**a**) turbulent kinetic energy *TKE*, and (**b**) spatial velocity gradient *SVG*. The green line indicates the position of the FGS in the model.

### 3.2.3. Hydraulic Parameters at the Bypass Entrance

Figure 5b shows the flow field at the bypass entrance at mid-flow depth ($z = 2.0$ m, $z/h_0 = 0.5$). The flow around the weir-side part of the dividing pier becomes relatively fast at the upstream end, similar to the turbine-side part described in Section 3.2.1, with a local velocity maximum of $v_{m,max} = 1.14$ m/s at $z = 3.5$ m ($z/h_0 = 0.875$, Figure 8b). From there in the direction of the bypass, a favorable flow direction occurs, as can be seen using the velocity vectors in Figure 5b. However, the velocities first decelerate before they slowly accelerate again in front of the inlet gate. The flow around the weir-side part of the dividing pier leads to a local velocity minimum at the upstream end of the turbine-side part and at the downstream end of the FGS, respectively, where the water flows directly onto the dividing pier. Moreover, this results in more turbulent flow conditions ($TKE_{max} = 0.07$ m$^2$/s$^2$ at $z = 2.0$ m, $z/h_0 = 0.5$, Figure 7a), and in increased $SVG$ above 2.0 cm/s/cm (Figure 7b). However, fish swimming from the weir around the weir-side part of the dividing pier towards the bypass also experience more complex flows with $TKE_{max} = 0.05$ m$^2$/s$^2$ and $SVG_{max} = 1.8$ cm/s/cm at $z = 2.0$ m ($z/h_0 = 0.5$). Overall, a continuous increase of velocities into the bypass as well as low $TKE$ and $SVG$, as recommended in common guidelines, are not present for fish migrating along the FGS nor for those approaching from the weir.

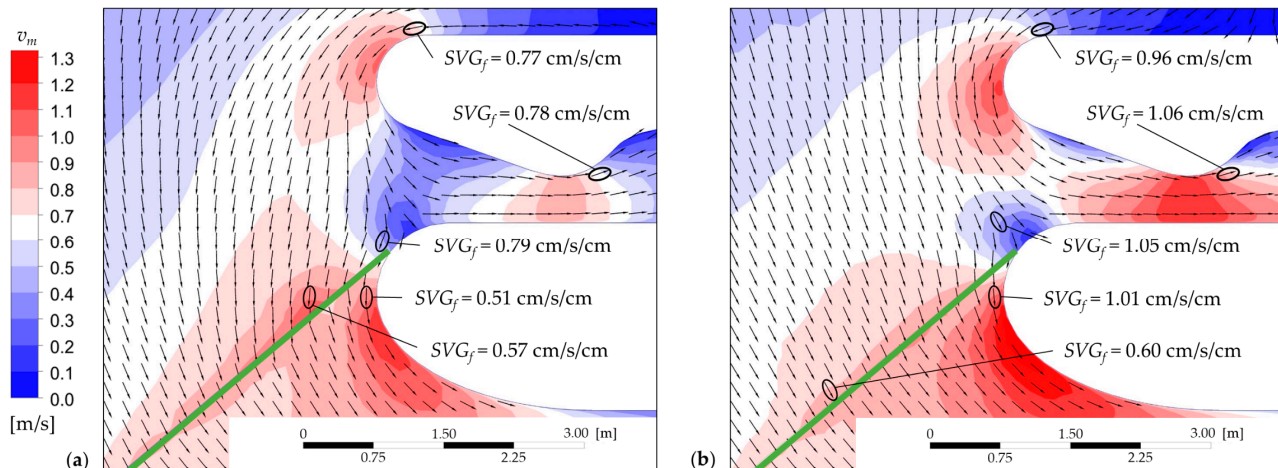

**Figure 8.** Velocity field at the bypass entrance of the HPP on the pre-alpine river (**a**) near the riverbed ($z = 0.1$ m, $z/h_0 = 0.025$) and (**b**) near the water surface ($z = 3.5$ m, $z/h_0 = 0.875$), including normalized flow vectors with a length of 0.2 m in a rectangular grid of 0.2 m distance and selected values for the spatial velocity gradient experienced by a fish $SVG_f$ with total length $TL = 0.2$ m. The green line indicates the position of the FGS in the model.

The flow field shows basically similar patterns in all three considered flow depths. The greatest difference occurs at the bypass entrance, where the FGE into the bypass further deteriorates near the riverbed ($z = 0.1$ m, $z/h_0 = 0.025$) due to a local velocity minimum over the whole width of the bypass entrance ($v_{m,max} = 0.37$ m/s, Figure 8a). In contrast, at mid-flow depth ($z = 2.0$ m, $z/h_0 = 0.5$, Figure 5b) and near the water surface ($z = 3.5$ m, $z/h_0 = 0.875$, Figure 8b), higher velocities occur in this area ($v_{m,max} = 0.68$ m/s at both water levels). This effect can be attributed to some extent to the sloping weir, which decelerates the inflow into the bypass near the riverbed ($z = 0.1$ m, $z/h_0 = 0.025$). Consequently, flow velocities increase to a local maximum close to the downstream end of the FGS at the same water level, as shown in Figure 6a.

Besides the velocity field, Figures 5b and 8 show selected values for $SVG_f$, which were calculated based on the normalized velocity vectors with a length of 0.2 m in the rectangular grid of 0.2 m distance and Equation (4). Note that $SVG_f$ could not be calculated automatically by the software during the analysis process. Therefore, only the maximum values near the two parts of the dividing pier and the inlet gate as well as in front of the FGS are shown in Figures 5b and 8. In the area upstream of the FGS and at the bypass entrance,

$SVG_f$ exceeds the value of 1.0 cm/s/cm only at a flow depth of $z = 3.5$ m ($z/h_0 = 0.875$) at two vectors near the local velocity minimum at the turbine-side part of the dividing part and close to the inlet gate, with $SVG_f = 1.05$ cm/s/cm and $SVG_f = 1.06$ cm/s/cm, respectively (Figure 8b). Compared to Figure 7b, the values for $SVG$ are significantly higher than those of $SVG_f$.

### 3.3. Variations

#### 3.3.1. Variation 1 (V1): Shifting the Weir-Side Part of the Dividing Pier 1.0 m in the Upstream Direction

In V1, the weir-side part of the dividing pier was shifted 1.0 m upstream at the HPP on the pre-alpine river. Besides that, no geometric and hydraulic parameters were changed. Figures 9 and 10 show the results (velocities) of V1 compared to the initial design. In V1, the flow to the downstream end of the FGS is more favorable compared to the initial design, with $\theta \leq 90°$ at mid-flow depth ($z = 2.0$ m, $z/h_0 = 0.5$, Figure 9), resulting in consistently positive values for $v_p$ (Figure 10d). However, negative values for $v_p$ still occur near the riverbed and near the water surface in this region, with $v_{p,min} = -0.42$ m/s at $z = 0.04$ m ($z/h_0 = 0.01$, Figure 10d). Furthermore, V1 only slightly affects $v_a$ and $v_n$ (Figure 10b) in front of the FGS, resulting in no significant improvement related to the ratio $v_p/v_n$. At the bypass entrance, the velocities increase due to the geometric variation, especially at the weir-side part of the dividing pier (from $v_{m,max} = 1.14$ m/s to $v_{m,max} = 1.36$ m/s, increase of 19.3%, at $z = 3.5$ m, $z/h_0 = 0.875$) and close to the inlet gate (from $v_{m,max} = 1.17$ m/s to $v_{m,max} = 1.20$ m/s, increase of 2.6%, at $z = 3.5$ m, $z/h_0 = 0.875$). Moreover, the velocities increase in the area of low velocities at the turbine-side part of the dividing pier, described in Section 3.2.3, with this area being shifted slightly towards the downstream end of the FGS compared to the initial design (Figure 5b). The increased velocities also increase $TKE$ (from $TKE_{max} = 0.07$ m$^2$/s$^2$ to $TKE_{max} = 0.09$ m$^2$/s$^2$, increase of 24.3%, at $z = 2.0$ m, $z/h_0 = 0.5$) at the bypass entrance as well as $SVG$, which does not decrease below $SVG = 1$ cm/s/cm over the whole entrance width near the inlet gate. Similarly, $SVG_f$ increases, involving two vectors directly adjacent to $SVG_f > 1$ cm/s/cm upstream of the FGS (Figures 9 and 10a). Overall, V1 increases $v_p$ towards the bypass and thus the FGE into the bypass, which could lead to fish finding the bypass more easily and migrating downstream of the HPP more quickly. However, the increased velocities and more complex flow conditions may also increase the probability of avoidance reactions.

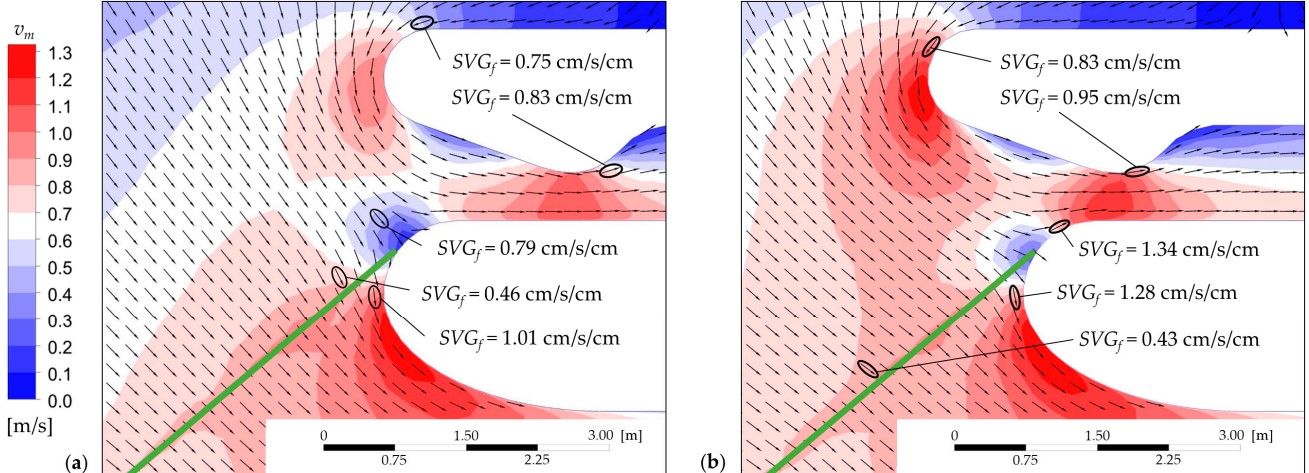

**Figure 9.** Velocity field at the bypass entrance of the HPP on the pre-alpine river at $z = 2.0$ m ($z/h_0 = 0.5$), including normalized flow vectors with a length of 0.2 m in a rectangular grid of 0.2 m distance and selected values for the spatial velocity gradient experienced by a fish $SVG_f$ with total length $TL = 0.2$ m for (**a**) the initial design, and (**b**) variation 1 (V1). The green line indicates the position of the FGS in the model.

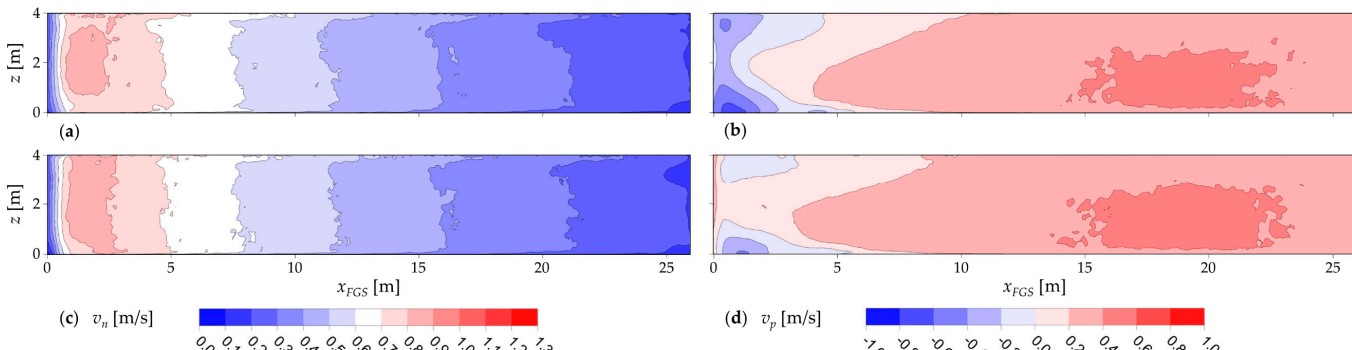

**Figure 10.** Velocities at the HPP on the pre-alpine river for (**a**,**b**) the initial design, and (**c**,**d**) variation 1 (V1) 0.1 m in front of the FGS: (**a**,**c**) rack normal velocity component $v_n$, and (**b**,**d**) rack parallel velocity component $v_p$ (positive in the bypass direction, negative in the direction of the upstream end of the FGS).

### 3.3.2. Variation 2 (V2): Changing the Shape and Width of the Weir-Side Part of the Dividing Pier

In order to achieve a smoother flow around the weir-side part of the dividing pier at the HPP on the pre-alpine river, both the shape and the width of this part were changed in V2. Here, the same geometry was used as for the turbine-side part with a width of 2 m but mirrored in plan view, and the inlet gate was placed in the same position as in the initial design (Table 2). Besides this, no further modifications were made. Figure 11 shows the velocity field of V2 at the bypass entrance at $z = 3.5$ m ($z/h_0 = 0.875$) in comparison with the initial design. As expected, the flow around the weir-side part of the dividing pier is slower and more uniform in V2, with $v_{m,max} = 1.14$ m/s and $SVG_{f,max} = 0.96$ cm/s/cm at $z = 3.5$ m ($z/h_0 = 0.875$) in the initial design and $v_{m,max} = 0.96$ m/s (decrease of 15.8%) and $SVG_{f,max} = 0.67$ cm/s/cm (decrease of 30.2%) at the same water level in V2. However, except for this area, V2 hardly affects the general flow field. Close to the inlet gate and in the area of low velocities at the turbine-side part of the dividing pier, velocities decrease marginally (Figure 11), as does *TKE*. Furthermore, the effect of this variation on the flow conditions at the FGS is negligible. Overall, V2 slightly improves the FGE due to lower velocities and more uniform flow, especially for fish approaching the bypass from the area of the weir.

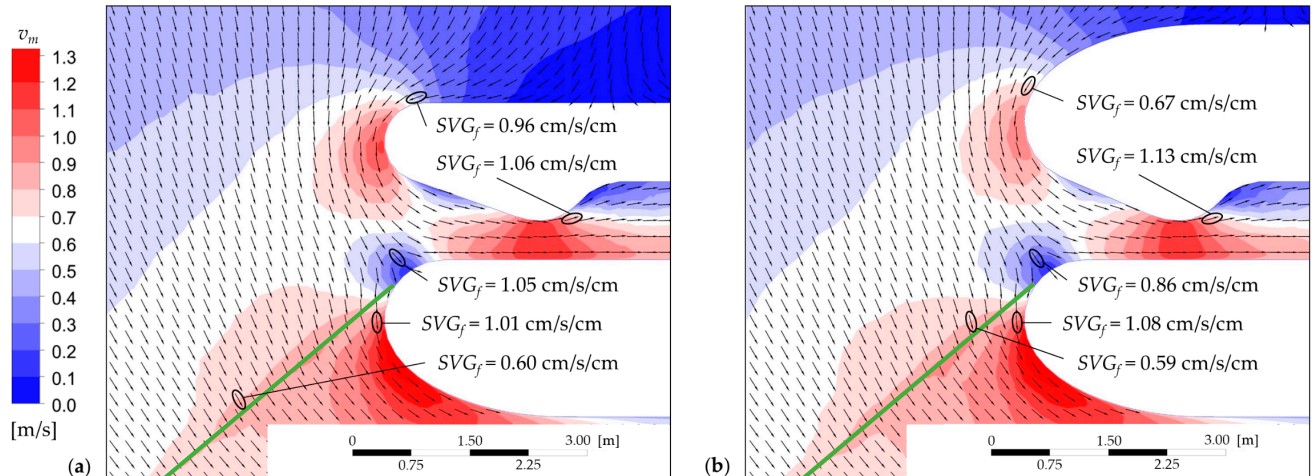

**Figure 11.** Velocity field at the bypass entrance of the HPP on the pre-alpine river at $z = 3.5$ m ($z/h_0 = 0.875$), including normalized flow vectors with a length of 0.2 m in a rectangular grid of 0.2 m distance and selected values for the spatial velocity gradient experienced by a fish $SVG_f$ with total length $TL = 0.2$ m for (**a**) the initial design, and (**b**) variation 2 (V2). The green line indicates the position of the FGS in the model.

### 3.3.3. Variation 3 (V3): Installing the Inlet Gate at the Turbine-Side Part of the Dividing Pier

V3 involved examining the effect of the position of the inlet gate on the flow field. For this, the inlet gate was installed at the turbine-side part of the dividing pier. While the flow field along the FGS is negligibly affected by this geometric variation, the flow coming from the area of the weir is further deflected in the direction of the bypass (Figure 12). In addition to increased velocities at the weir-side part of the dividing pier (from $v_{m,max}$ = 1.14 m/s to $v_{m,max}$ = 1.16 m/s, increase of 1.8%, at $z$ = 3.5 m, $z/h_0$ = 0.875) and close to the inlet gate (from $v_{m,max}$ = 1.17 m/s to $v_{m,max}$ = 1.37 m/s, increase of 17.1%, at $z$ = 3.5 m, $z/h_0$ = 0.875), a large-scale increase of *TKE* from the bypass entrance until shortly after the constriction at the inlet gate occurs. However, the maximum value of *TKE* remains at $TKE_{max}$ = 0.07 m²/s² ($z$ = 2.0 m, $z/h_0$ = 0.5). Moreover, *SVG* also increases slightly in this area. Overall, V3 leads to larger flow deflections and more complex flow conditions, which may increase the probability that fish swimming towards the bypass show an avoidance reaction.

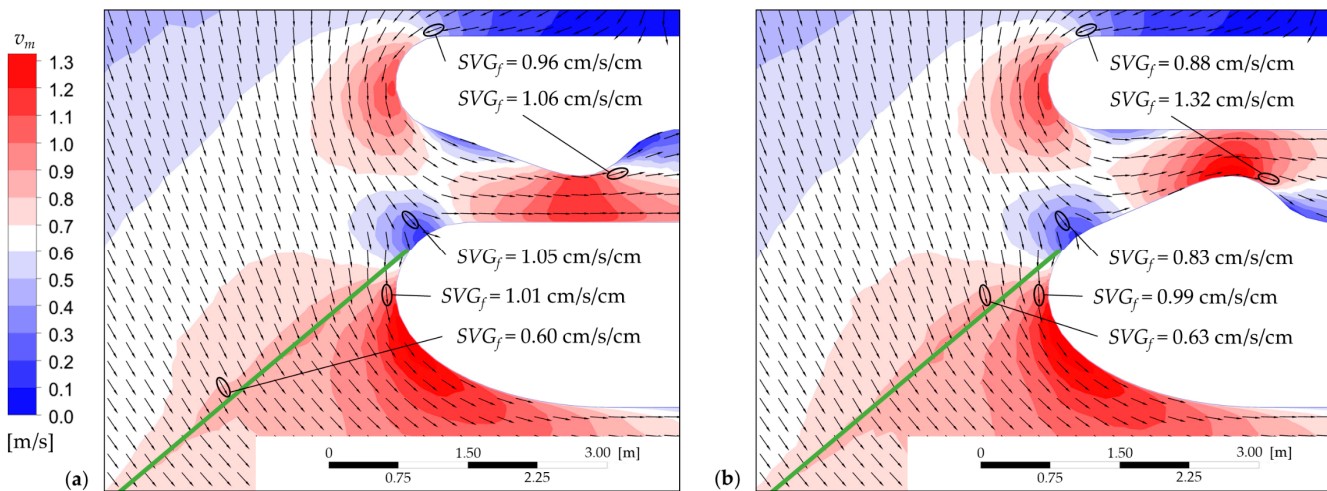

**Figure 12.** Velocity field at the bypass entrance of the HPP on the pre-alpine river at $z$ = 3.5 m ($z/h_0$ = 0.875), including normalized flow vectors with a length of 0.2 m in a rectangular grid of 0.2 m distance and selected values for the spatial velocity gradient experienced by a fish $SVG_f$ with total length $TL$ = 0.2 m for (**a**) the initial design, and (**b**) variation 3 (V3). The green line indicates the position of the FGS in the model.

### 3.3.4. Variation 4 (V4): Doubling $Q_{by}$ by Lowering the Sloping Weir

In V4, and further in V5, $Q_{by}$ was increased by reducing the height of the sloping weir. For both variations, the HPP on the alpine river was used as the initial design due to lower velocities at the bypass entrance (Figure 13a) compared to the HPP on the pre-alpine river (Figure 5b). Additionally, the initial design shows unfavorable backflow effects and partly high $SVG_f$ ($SVG_{f,max}$ = 1.85 cm/s/cm at $z$ = 1.0, $z/h_0$ = 0.5) due to vortex formation at the bypass entrance near the weir-side part of the dividing pier (Figure 13a). It should be noted that for the calculation of $SVG_f$ at the HPP on the alpine river, $TL$ = 0.1 m was assumed, while $TL$ = 0.2 m was used for the HPP on the pre-alpine river. In V4, $Q_{by}$ was doubled compared to the initial design (from $Q_{by}$ = 0.5 m³/s, 5% of $Q_0$, to $Q_{by}$ = 1.0 m³/s, 10% of $Q_0$). Since $Q_0$ remains constant, $Q_d$ decreases and thus also $v_n$ towards the FGS (from $v_{n,max}$ = 0.78 m/s to $v_{n,max}$ = 0.75 m/s, decrease of 3.2%, 0.1 m in front of the FGS). Moreover, the FGE along the FGS is slightly improved, with higher $v_p$, although negative values for $v_p$ continue to occur at the downstream end of the FGS (from $v_{p,min}$ = −0.59 m/s to $v_{p,min}$ = −0.34 m/s, increase of 42.4%, 0.1 m in front of the FGS). Figure 13b shows the velocity field of V4 at the bypass entrance in mid-flow depth ($z$ = 1.0, $z/h_0$ = 0.5). While in the area where the FGS is fixed to the turbine-side part of the dividing pier, the velocities are still relatively low, and a flow deceleration occurs, a consistent increase in velocity can be observed from the bypass entrance to the inlet gate. However, the flow velocities around

the weir-side part of the dividing pier increase (from $v_{m,max} = 0.97$ m/s to $v_{m,max} = 1.23$ m/s, increase of 26.8%, at $z = 1.0$, $z/h_0 = 0.5$), and further also $SVG_{f,max}$ (from 1.35 cm/s/cm to 1.71 cm/s/cm, increase of 26.7%, at $z = 1.0$, $z/h_0 = 0.5$). Accordingly, more complex flow conditions with increasing values for *TKE* and *SVG* occur in this area. Overall, it can nevertheless be assumed that V4 has a positive effect on the FGE both along the FGS and at the bypass entrance.

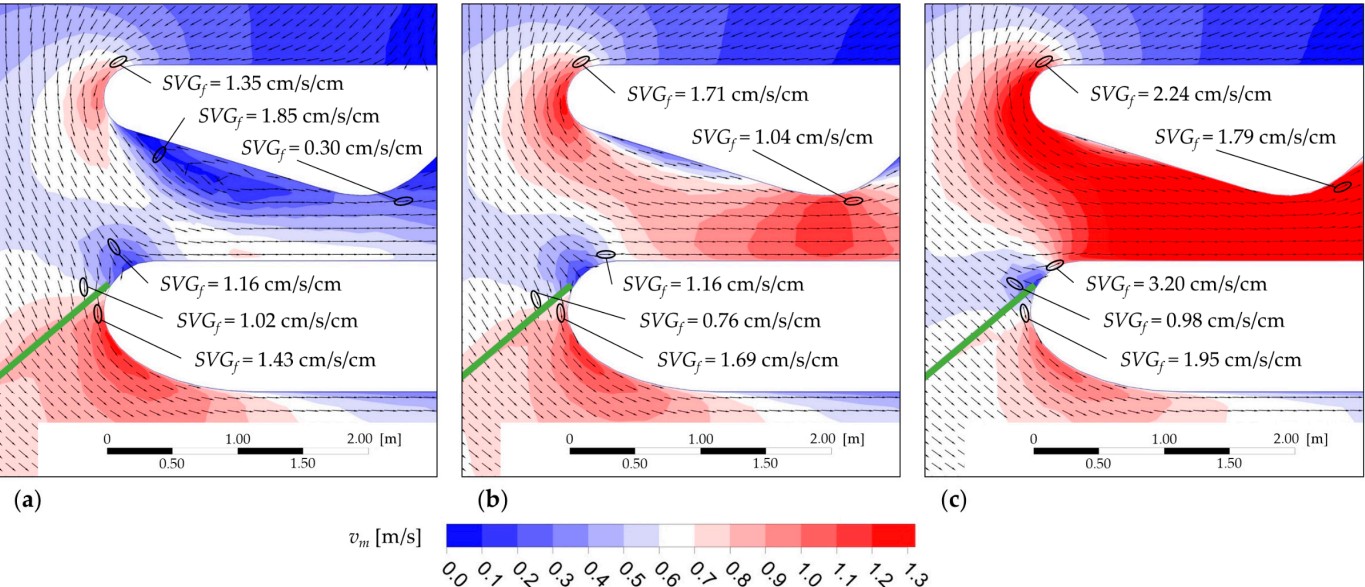

**Figure 13.** Velocity field at the bypass entrance of the HPP on the alpine river at $z = 1.0$ m ($z/h_0 = 0.5$), including normalized flow vectors with a length of 0.1 m in a rectangular grid of 0.1 m distance and selected values for the spatial velocity gradient experienced by a fish $SVG_f$ with total length $TL = 0.1$ m for (**a**) the initial design, (**b**) variation 4 (V4), and (**c**) variation 5 (V5). The green line indicates the position of the FGS in the model.

### 3.3.5. Variation 5 (V5): Quadrupling $Q_{by}$ by Lowering the Sloping Weir

Similar to V4, $Q_{by}$ was doubled in V5 compared to V4 and quadrupled compared to the initial design, respectively ($Q_{by} = 2.0$ m³/s, 20% of $Q_0$). At the FGS, basically the same tendencies are evident in V5 as in V4, with decreasing $v_a$ and $v_n$, as well as increasing $v_p$ at the downstream end. Except for a small area near the water surface at the downstream end, the values for $v_p$ along the FGS are continuously positive in V5. Nevertheless, the ratio $v_p/v_n$ does not increase above the value of 1 in the downstream half of the FGS. At the bypass entrance, the flow velocities increase significantly (Figure 13c) compared to the initial design, such as at the weir-side part of the dividing pier (from $v_{m,max} = 0.97$ m/s to $v_{m,max} = 1.70$ m/s, increase of 75.3%, at $z = 1.0$ m, $z/h_0 = 0.5$) and close to the constriction of the inlet gate (from $v_{m,max} = 0.74$ m/s to $v_{m,max} = 2.60$ m/s, increase of 251.4%, at $z = 1.0$ m, $z/h_0 = 0.5$). In addition, the flow conditions become more complex, e.g., with $SVG_{f,max} = 3.20$ cm/s/cm at $z = 1.0$ m ($z/h_0 = 0.5$) near the turbine-side part of the dividing pier at the bypass entrance (Figure 13c). Overall, despite the improved FGE along the FGS, V5 indicates very high flow velocities and more complex flow conditions at the bypass entrance, and therefore it can be assumed that fish show avoidance reactions when swimming into this area.

### 3.3.6. Variation 6 (V6): Changing $\alpha$ to 20°

In V6, $\alpha$ was changed from 40° to 20° at the HPP on the alpine river. Consequently, $l_{FGS}$ increased from 10.58 m to 20.19 m. Similar to the initial design, the FGS was attached tangentially to the upstream end of the turbine-side part of the dividing pier, which altered the location of the FGS in the model. Figure 14 shows the velocities $v_n$ and $v_p$ 0.1 m in front

of the FGS for V6 compared to the initial design, and Figure 15 shows the velocity field at the downstream end of the FGS and the bypass entrance in mid-flow depth ($z$ = 1.0 m, $z/h_0$ = 0.5). Due to the larger $A_{FGS,hyd}$ in V6, $\overline{v_n}$ decreases from 0.45 m/s to 0.24 m/s (decrease of 46.7%). Moreover, $v_{n,max}$ decreases from 0.78 m/s to 0.70 m/s (decrease of 10.3%, 0.1 m in front of the FGS, Figure 14a,c). However, since the FGS has only a minor effect on the flow conditions at its downstream end as described in Section 3.2.2, the decrease of $v_{n,max}$ can be attributed primarily to the new location of the FGS in the model (Figure 15). Furthermore, with decreasing $\alpha$, $\theta$ also decreases, resulting in almost consistently positive values for $v_p$ along the FGS (Figure 14d). Nevertheless, negative values for $v_p$ still occur at the downstream end (from $v_{p,min}$ = −0.60 m/s to $v_{p,min}$ = −0.29 m/s, increase of 51.7%, 0.1 m in front of the FGS), implying that no guiding effect in the bypass direction appears in this region. The ratio $v_p/v_n$ has a value above 1 over more than half of $l_{FGS}$, which is favorable for the FGE. As expected, due to the minor effect of the FGS on the flow, the flow field hardly changes at the bypass entrance (Figure 15). Overall, V6 has a positive effect on the FGE from a hydraulic perspective and based on the evaluation criteria defined in Section 2.3.

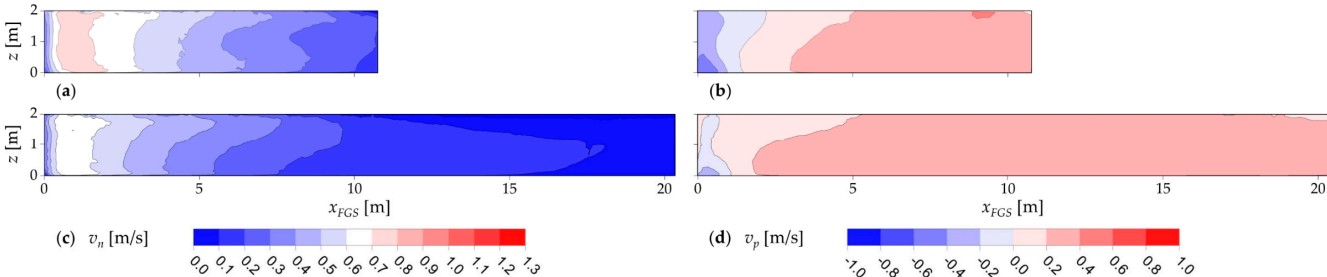

**Figure 14.** Velocities at the HPP on the alpine river for (**a**,**b**) the initial design, and (**c**,**d**) variation 6 (V6) 0.1 m in front of the FGS: (**a**,**c**) rack normal velocity component $v_n$, and (**b**,**d**) rack parallel velocity component $v_p$ (positive in the bypass direction, negative in the direction of the upstream end of the FGS).

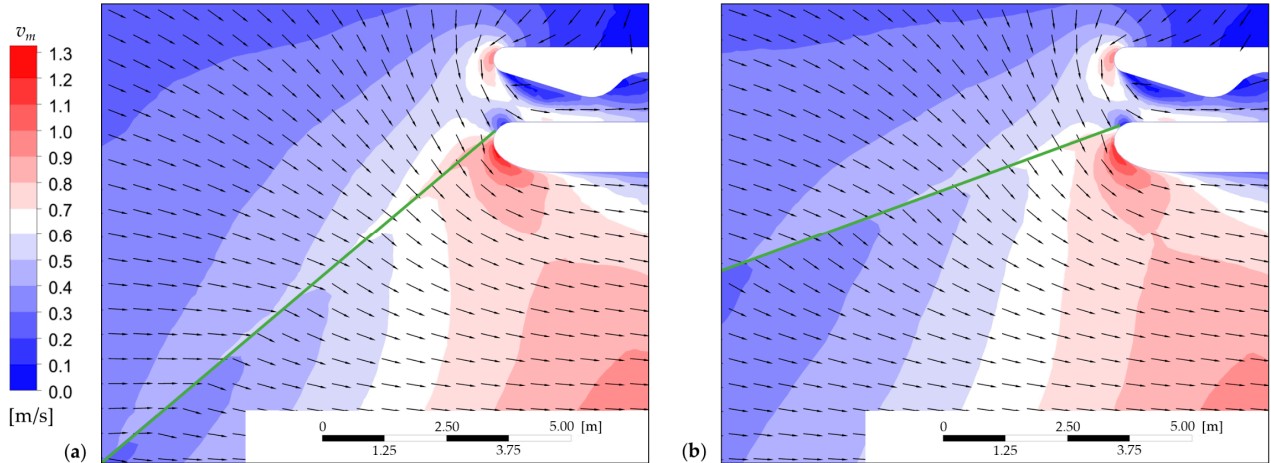

**Figure 15.** Velocity field at the HPP on the alpine river at $z$ = 1.0 m ($z/h_0$ = 0.5): velocity magnitude $v_m$ and normalized flow vectors with a length of 0.5 m in a rectangular grid of 0.5 m distance around the FGS for (**a**) the initial design, and (**b**) variation 6 (V6). The green line indicates the position of the FGS in the model.

3.3.7. Variation 7 (V7): Implementing a Bottom Overlay with a Height of 0.2 m

A bottom overlay with a height of $h_{bo}$ = 0.2 m ($h_{bo}/h_0$ = 0.1) was applied to the FGS of the HPP on the alpine river in V7. The bottom overlay is represented in the numerical model as an impermeable body with the same thickness as the FGS, and thus reduces $A_{FGS,hyd}$ by 10%. In V7, further flow deflections occur in front of the FGS towards its

upstream end, particularly near the riverbed at $z = 0.1$ m ($z/h_0 = 0.05$, Figure 16). At this flow depth, $v_n$ decreases significantly (Figure 17a,c), while $v_p$ hardly changes compared to the initial design (Figure 17b,d). For the ratio $v_p/v_n$, values above 1 occur in the upstream half of the FGS, but remain below 1 in the downstream half, even though $v_n$ is close to 0. This is due to the fact that the flow vectors are aligned against the bypass direction (Figure 16). Moreover, V7 also leads to more complex flow conditions with increased *TKE* and *SVG* near the riverbed ($z = 0.1$ m, $z/h_0 = 0.05$). The latter can be related to accelerations and decelerations caused by the bottom overlay. In mid-flow depth ($z = 1.0$ m, $z/h_0 = 0.5$) and near the water surface ($z = 1.5$ m, $z/h_0 = 0.75$), the bottom overlay has a small effect on the flow field. While $v_{n,max}$ hardly changes due to the variation ($v_{n,max} \approx 0.78$ m/s, 0.1 m in front of the FGS), marginally higher $v_n$ and lower $v_p$ occur in the $x_{FGS}$-direction (Figure 17). At the bypass entrance, the effect of the bottom overlay is low, with the largest differences near the riverbed at $z = 0.1$ m ($z/h_0 = 0.05$, Figure 16). It should be noted that the bottom overlay also affects the flow field downstream of the FGS, particularly with significant flow deflections parallel to the FGS near the riverbed at $z = 0.1$ m ($z/h_0 = 0.05$, Figure 16). However, this area is not part of the present study. Overall, the flow vectors close to the bottom overlay at $z = 0.1$ m ($z/h_0 = 0.05$) are strongly deflected in the upstream direction, which can lead to a deterioration of the findability of the bypass and to delays for fish migrating downstream. Nevertheless, fish are effectively protected from impingement on the FGS due to low $v_n$ towards the bottom overlay, which are lower than $v_{sus}$ even for small fish.

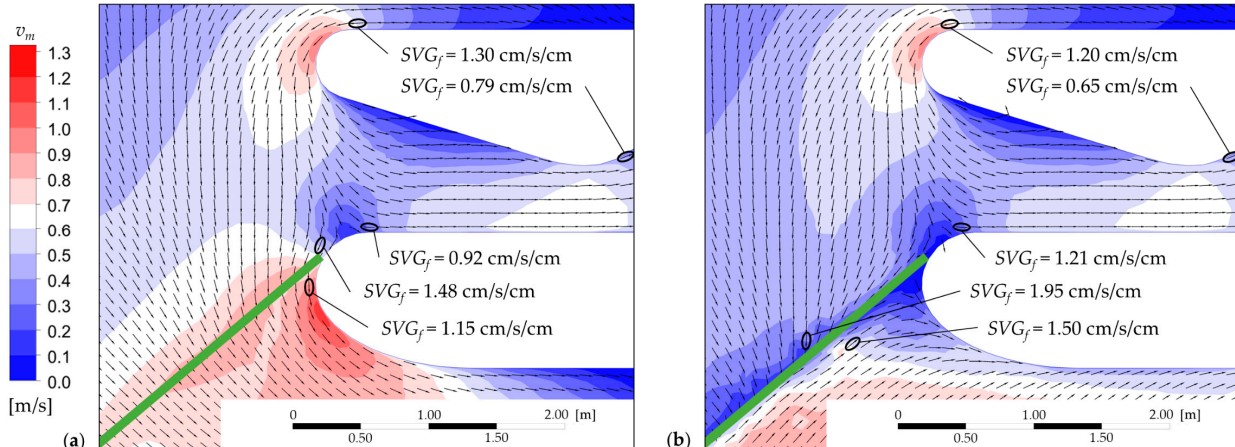

**Figure 16.** Velocity field at the downstream end of the FGS and bypass entrance of the HPP on the alpine river at $z = 0.1$ m ($z/h_0 = 0.05$), including normalized flow vectors with a length of 0.1 m in a rectangular grid of 0.1 m distance and selected values for the spatial velocity gradient experienced by a fish $SVG_f$ with total length $TL = 0.1$ m for (**a**) the initial design, and (**b**) variation 7 (V7). The green line indicates the position of the FGS in the model.

3.3.8. Variation 8 (V8): Integrating the FGS into the Headrace Channel with the Bypass on the Orographic Right Side

To examine whether a solution with the FGS installed in the headrace channel would be appropriate for the HPP designs presented, V8 was conducted for the HPP on the pre-alpine river. Here, the upstream end of the FGS was installed near the upstream end of the dividing pier, and the bypass entrance was constructed in front of the turbine inlets in the headrace channel on the orographic right bank. Note that the headrace channel in V8 was extended by 5 m in the downstream direction to allow the FGS ($\alpha = 40°$, $l_{FGS} = 23.34$ m) to be integrated into the headrace channel in front of the inclined concrete bottom. Figure 18 shows the velocity field of V8 in mid-flow depth ($z = 2.0$ m, $z/h_0 = 0.5$). In V8, the flow around the dividing pier is relatively fast ($v_{m,max} = 1.78$ m/s, $z = 3.5$ m, $z/h_0 = 0.875$), and more complex flow conditions with $TKE_{max} = 0.07$ m²/s² at $z = 2.0$ m ($z/h_0 = 0.5$) appear. In the headrace channel, high velocities ($v_m \geq 0.9$ m/s) occur upstream of the

FGS. However, the approach flow to the FGS is favorable with $\theta \leq 40°$ along the FGS. As a result, the values of $v_n$ are relatively evenly distributed over $A_{FGS,hyd}$ 0.1 m in front of the FGS ($v_{n,max} = 0.74$ m/s), and consistently positive values occur for $v_p$ ($v_{p,max} = 1.07$ m/s). Consequently, the ratio $v_p/v_n$ is also consistently above the value 1. At the bypass entrance, the velocities are slightly reduced, partly due to the inlet gate, which was not considered in V8 due to high velocities along the FGS. Moreover, this also leads to increased $SVG$ and $SVG_f$ in this area compared to the main headrace channel, with $SVG_{f,max} = 0.72$ cm/s/cm upstream of the FGS at $z = 2.0$ m ($z/h_0 = 0.5$, Figure 18b). In addition, backflow effects occur at the bypass entrance near the riverbed ($z = 0.1$ m, $z/h_0 = 0.025$), which can be attributed to the sloping weir located relatively close to the bypass entrance. Overall, although V8 shows favorable flow conditions regarding $v_p$, $v_n$ and $v_p/v_n$ in front of the FGS, high flow velocities ($v_m \geq 0.9$ m/s) occur upstream of the FGS in the headrace channel, which are above $v_{pro}$ for fish with $TL \leq 0.12$ m (Figure 4). A constant velocity increase into the bypass, as recommended by guidelines, would increase these velocities even further.

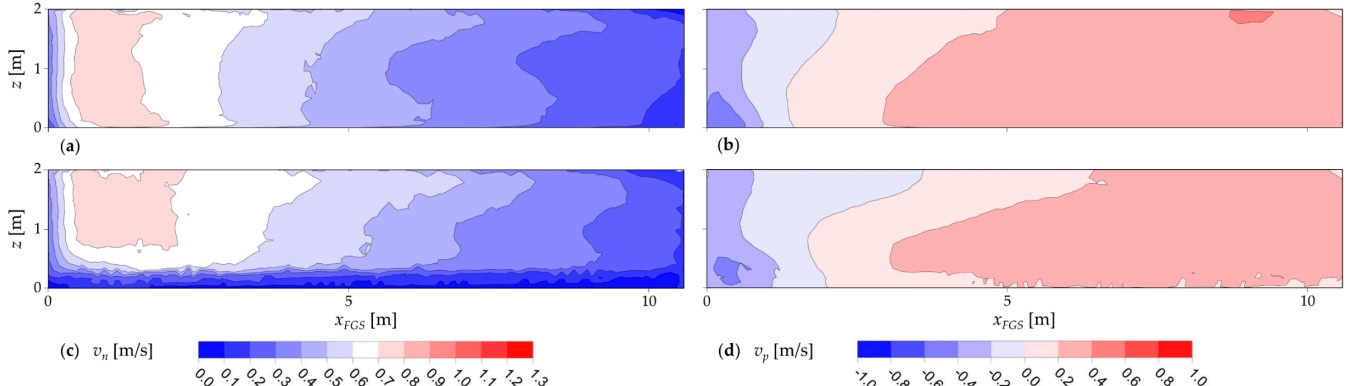

**Figure 17.** Velocities at the HPP on the alpine river for (**a**,**b**) the initial design, and (**c**,**d**) variation 7 (V7) 0.1 m in front of the FGS: (**a**,**c**) rack normal velocity component $v_n$, and (**b**,**d**) rack parallel velocity component $v_p$ (positive in the bypass direction, negative in the direction of the upstream end of the FGS).

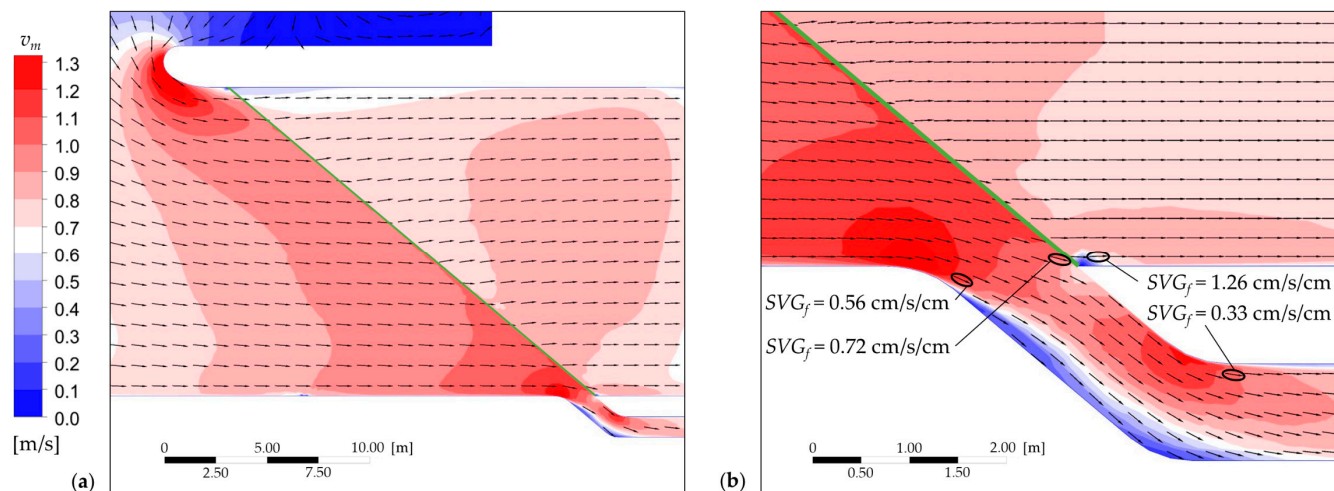

**Figure 18.** Velocity field at the HPP on the pre-alpine river for variation 8 (V8) at $z = 2.0$ m ($z/h_0 = 0.5$): (**a**) velocity magnitude $v_m$ and normalized flow vectors with a length of 0.75 m in a rectangular grid of 1 m distance around the FGS, and (**b**) velocity magnitude $v_m$, normalized flow vectors with a length of 0.2 m in a rectangular grid of 0.2 m distance and selected values for the spatial velocity gradient experienced by a fish $SVG_f$ with total length $TL = 0.2$ m at the bypass entrance. The green line indicates the position of the FGS in the model.

## 4. Discussion

### 4.1. Interpretation of the Results and Comparison with the Literature

Previous studies have predominantly investigated HBRs with relatively homogenous inflow conditions, either in field measurements on diversion HPPs (e.g., [25,39]) or in laboratory tests in flumes (e.g., [30,36,80,81]). In contrast, to the authors' knowledge, the block-type layout with an FGS has only been studied by Meister et al. [41] in laboratory hydraulic tests, but without considering the bypass. In the present study, the results of the numerical simulations of the initial designs show that the flow conditions upstream of FGSs and at the bypass entrance of block-type HPPs are less favorable than the generally relatively homogenous flow conditions at similar facilities in headrace channels of diversion HPPs. While only low flow velocities occur at the downstream end of the FGS in the initial design of the HPP on the pre-alpine river, $v_{n,max}$ = 0.87 m/s significantly exceeds $\overline{v_n}$ = 0.46 m/s by 89.1% at the upstream end (Section 3.2.2). Therefore, using simplified averaged values, as frequently adopted in common guidelines, can result in adverse conditions for downstream migrating fish by ignoring the spatial deviation of the flow field, as well as the temporal deviation [82], which, however, was not examined in the present study. Similar unfavorable velocity distributions upstream of an FGS were also observed by Berger [80] at an HPP with an HBR. Comparing Figures 4–6, the criterion $v_{n,max} \leq v_{sus}$ is met for fish with $TL \geq 0.22$ m, and $v_{n,max} \leq v_{pro}$ for fish with $TL \geq 0.12$ m. Maddahi et al. [39] compared calculated values for $v_{sus}$ and $v_{pro}$ with video and ARIS Sonar monitoring results of fish upstream of an HBR-BS at a diversion HPP, and observed that fish can swim against the flow in front of the HBR even at $v_n$ between $v_{sus}$ and $v_{pro}$. Thus, the use of $v_n \leq v_{sus}$ represents the conservative approach. A high proportion of downstream migrating fish consists of small species or small individuals [83]. Geist [84] assumed that the majority of fish (typically > 90%) in European streams and rivers are smaller than $TL$ = 0.15 m. This indicates that in worst case scenarios (worst case operating conditions, fish swimming in or close to areas with $v_{n,max}$), risks can occur for most fish at the HPP on the pre-alpine river. Furthermore, the results of the numerical simulations of the initial designs showed that the approach flow to the downstream end of the FGS is perpendicular or even slightly deflected against the bypass direction, which is comparable to the results of Meister et al. [41] for block-type layouts. A suitable FGE with $v_p/v_n > 1$ occurs only in the upstream half of the FGS. In this regard, it should be noted that the approach conditions to HBRs are not only important for fish protection; there are also operational aspects since, for instance, suitable values for $v_p$ are needed for automated rack cleaning at HBRs, or local head losses are increased with velocity quadratically and with $\sin(\theta)$ linearly (for $0° \leq \theta \leq 90°$) [37]. Besides the FGS, the flow conditions at the bypass entrance at the HPP on the pre-alpine river are not favorable either (Section 3.2.3).

Silva et al. [47] investigated downstream movements of Atlantic salmon smolts at an HPP in Norway by means of 3D numerical simulations, as well as 2D and 3D telemetry, and determined values for $TKE \leq 0.24$ m$^2$/s$^2$ in the intake area of the HPP, and $TKE \leq 0.03$ m$^2$/s$^2$ in the main river course. They concluded that, for $TKE$ between 0.03 and 0.24 m$^2$/s$^2$, the swimming performance of fish was affected, while for $TKE < 0.03$ m$^2$/s$^2$ it was positively influenced. Liao [48] confirmed that low turbulent flows, which do not pose a threat, can be attractive for fish. He concluded that there is a relation between fish size and turbulence strength in which fish like to remain [48]. Other values at which different fish species respond to increased $TKE$ can be found in the literature; e.g., Silva et al. [85] showed that barbel are adequately adapted for $TKE < 0.05$ m$^2$/s$^2$, while Li et al. [86] indicated that juvenile cyprinids with $TL \approx 0.11$ m may react to $TKE < 0.005$ m$^2$/s$^2$. Even native fishes can respond differently to turbulent flows, as shown by Link et al. [87] for two Chilean native fish species. Furthermore, Szabo-Meszaros et al. [81] determined values for $TKE$ between 0.01 and 0.04 m$^2$/s$^2$ in laboratory experiments at the bypass entrance, and $TKE$ smaller than $\approx 0.005$ m$^2$/s$^2$ upstream of the HBR. Even lower values for $TKE$ upstream of the HBR were found by Meister et al. [40]. The results of the HPP on the pre-alpine river in this study show that $TKE$ is between 0.04 and 0.07 m$^2$/s$^2$ over the entire width of the bypass

entrance (Figure 7a), and in front of the FGS mostly $< 0.01$ m$^2$/s$^2$, except for the area close to the downstream end. Therefore, the computed values for *TKE* seem plausible compared to previous studies. However, a prediction of the possible fish behavior is hardly possible due to the lack of data [31,34,48], especially for the relevant fish species in the grayling and lower trout regions. Nevertheless, it cannot be excluded that values for *TKE* between 0.04 and 0.07 m$^2$/s$^2$ could result in impairments of swimming performance, or fish could show avoidance responses. Further studies with the relevant fish species are needed to validate this assessment.

Likewise, there are few data in the literature related to *SVG* [12]. Enders et al. [76] observed in laboratory experiments that Chinook salmon smolts exhibited avoidance responses at *SVG* between 1.0 and 1.2 cm/s/cm (both deceleration and acceleration), which agrees with the results of Haro et al. [45] for Atlantic salmon smolts and juvenile American shad. In Enders et al. [12], avoidance responses were monitored for the same *SVG* but varying discharges. Based on this, Vowles and Kemp [50] concluded that once a certain value of *SVG* occurs, fish exhibit an avoidance response. Moreover, in their ethohydraulic laboratory experiments, brown trout responded to *SVG* starting at $\approx 0.4$ cm/s/cm independent of the discharge, and under light conditions during nighttime experiments at $\approx 0.2$ cm/s/cm. In contrast, Boes et al. [43] observed rapid bypass acceptance for spirlin at *SVG* $\approx 0.6$ cm/s/cm. In this study, the results of the HPP on the pre-alpine river show that the highest values for *SVG* occur at the bypass entrance close to the walls, while a migration corridor with *SVG* $< 1.0$ cm/s/cm persists in the middle (Figure 7b). The values for $SVG_f$ are generally lower than those for *SVG*. In addition, the values for $SVG_f$ are lower at the HPP on the pre-alpine river compared to those at the HPP on the alpine river, which may be explained by the fact that TL = 0.1 m was assumed for the evaluation of the former, and TL = 0.2 m for the latter. It can be expected that the smaller the length of the vectors are, and the finer the rectangular mesh is resolved, the higher the values for $SVG_f$ increase until they reach the values of *SVG* at very fine resolutions. Further, the proposed approach to determine $SVG_f$ is highly position dependent; e.g., in the case of vortices, significantly different $SVG_f$ can occur depending on the position of the vectors considered (cf. Figure 13). However, it should be noted that some previous studies (e.g., [88]) also investigated the influence of acceleration on fish behavior, but not using *SVG*. Overall, more data is needed to make further conclusions about how fish react to abrupt acceleration and deceleration.

Generally, Silva et al. [47] suggested that for the evaluation of the effects of hydraulics on fish swimming performance, the interaction of hydraulic parameters should be considered as well. In the present study, the hydraulic parameters examined often behave independently; e.g., the highest values for *TKE* and *SVG* do not occur at the same locations in most cases. Nevertheless, high flow velocities increase the probability of appearing complex flows, as shown, for instance, at the bypass entrance of the initial design of the HPP on the pre-alpine river in Figures 5 and 7. From an ecological perspective, however, highly turbulent structures can have a destabilizing effect on fish, causing them to actively reduce their hydrodynamic resistance to stabilize, which reduces their maximum swimming speed [82]. This confirms the importance of studying the interaction of hydraulic parameters.

Based on the results of the two initial HPP designs related to downstream fish migration, V1 to V8 were performed. These variations should not be regarded as specific solutions to avoid unfavorable flow conditions. Instead, the effects of each variation on the flow field are shown in Section 3.3. In V1, the weir-side part of the dividing pier was shifted 1.0 m in the upstream direction, resulting in improved approach flows towards the bypass, although more complex flow conditions occur. However, it can be assumed that shifting the weir-side part further upstream is limited in order to not provide similar flow conditions as in V8. Another geometric variation of the weir-side part with positive effects on the flow field was performed in V2, indicating the importance of examining and optimizing the geometry and, based on V1, the position of the dividing pier at block-type HPPs regarding downstream fish migration. In V3, the inlet gate was installed at the turbine-side part of

the dividing pier. In principle, this follows the recommendation that the bypass entrance should be in line with the FGS to increase the findability of the bypass for fish [34], and thus avoid damage, delay, escape, or exhaustion [43]. For the block-type layout, however, this tends to deteriorate the flow field at the bypass entrance. V4 demonstrates the significance of an adequate bypass discharge, which should be defined independently of the recommended percentages for $Q_{by,rel}$, as shown by comparing the flow conditions at the bypass entrance of both initial designs (Figures 5b and 13a). This agrees with the recommendation of Boes et al. [43] to define $Q_{by}$ based on the hydraulic characteristics at the bypass entrance for an efficient bypass design. For the FGE along the FGS, the increase of $Q_{by}$ has a negligible effect. Additionally, V5 shows that the bypass discharge cannot be increased arbitrarily without creating adverse conditions for downstream migrating fish. Injuries or mortalities cannot be excluded if fish are unable to react quickly enough due to rapidly changing flow conditions. In the case of V5, it may be more beneficial to open a weir at least partially, thus providing another downstream migration opportunity, which is consistent with adding a spillway that can be beneficial for downstream migration at low head HPPs [89,90]. However, it should be noted that an additional downstream corridor may alter the flow conditions at the FGS and the bypass entrance [41]. In V6, $\alpha = 20°$ was defined for the HPP on the alpine river, which improves the hydraulic conditions in the area of interest as well as the FGE. For the HPP on the pre-alpine river, on the other hand, $\alpha = 20°$ would lead to $l_{FGS} = 49.31$ m, which significantly increases the average time required for fish to find the bypass as well as the risk of undesired passage through the FGS. In this regard, Nordlund [91] suggested that multiple bypass entrances should be used if $v_p$ is not guiding the fish to the bypass within 60 s. An opportunity to decrease $l_{FGS}$ and use lower $\alpha$ at the same time are HPPs in the bay-type layout, but it should be considered that the approach flow can be deflected even further compared to block-type HPPs [92]. Finally, technical and economic issues must also be considered when implementing longer FGSs, e.g., higher investment costs. The addition of a bottom overlay in V7 significantly decreases $v_n$ near the riverbed. In previous ethohydraulic model tests considering bottom overlays [40,77,93], an improved FGE was reported. This can be explained, since the majority of fish examined in model tests migrated near the riverbed; for instance, 97% of rack passages were observed close to the riverbed when no bottom overlay at an HBR was used in Meister et al. [40]. However, the protective function of bottom overlays has not yet been confirmed in field studies [94]. Areas of low flow velocities, such as in front of the bottom overlay, can also be used by fish to maintain position in the current without actively swimming, allowing them to rest [95]. Moreover, in the case of no pronounced guiding effect towards the bypass near the bottom overlay (i.e., $v_p \leq 0$ m/s), as shown close to the downstream end of the FGS in Figure 17, sediment deposition may occur at these locations, which can further deteriorate downstream fish migration, e.g., if delays occur as a result, increasing the probability of becoming a target for predators [35]. It should be mentioned that $h_{bo}$ used in V7 is lower than the recommended values in Ebel [28] ($h_{bo}/h_0 = 15$ to 20% or $h_{bo} \geq 0.5$ m). Nevertheless, it can be assumed that similar tendencies on the flow field occur at bottom overlays with $h_{bo} > 0.2$ m. V8 shows that for the defined HPP designs, installing the FGS upstream of the headrace channel should be the preferred option due to the velocities in the headrace channel being so high that small fish in particular may not be able to swim against the flow. Consequently, they only drift with the flow, which can cause them to be pressed against or through the FGS or enter areas with highly complex flow conditions. Therefore, fish should be prevented from entering the headrace channel in this case, and instead should be guided to a bypass further upstream. Nevertheless, since the hydraulic parameters in front of the FGS provides the best FGE of all designs studied, V8 may be considered for existing HPPs or new construction with low flow velocities in the headrace channel. In addition, a variant not included in this paper was examined in which the FGS was constructed in the headrace channel, and the bypass was integrated into the dividing pier. However, this variant leads to similar results as V8 and is therefore not presented in this study. In summary, it can be

expected that a combination of several variations, which also go beyond those presented in this study, can lead to significantly improved FGE at block-type HPPs.

### 4.2. Limitations

Several limitations should be considered, though, when interpreting the results of this study. First, the HPP designs studied are intended to be representative but simplified designs of existing block-type HPPs. Once the presented assessment is performed for a real HPP, site-specific conditions that may affect the flow field need to be considered, and in the case of an existing HPP, measurements to validate the results should be conducted, e.g., velocity measurements. Since the latter does not apply to the HPPs in this study, a model based on the HyTEC (Hydromorphology and Temperature Experimental Channel) facility in Lunz am See, Austria, where ethohydraulic experiments were previously conducted [44,96], was used for simplified validation. 3D numerical models of this facility were created, simulated, and the results subsequently compared to velocity measurements taken with Vectrino ADV and described in Haug [97]. The validated settings were subsequently applied to the numerical models in this study. Furthermore, the studied variants consider the associated advantages and disadvantages only in a simplified manner. A closer examination would not be expedient within the scope of this study but can be recommended for detailed planning. In addition, several assumptions were made during the modeling process. For example, all simulations used the most unfavorable flow conditions for downstream migrating fish (turbines in full operation, closed weir). However, it is important that migration facilities function over the entire flow range [16], not just the worst case.

Second, the focus of the present study is not on the design of the bypass, hence simplified assumptions were made that may partially influence the results. In particular, the sloping weir was placed immediately downstream of the inlet gate. It can be assumed that if it were positioned further downstream, more uniform flow conditions could be expected over the entire water height at the bypass entrance. This would explain the poor FGE observed near the riverbed ($z = 0.1$ m), which occurs frequently in the results (e.g., Figure 8a). In addition, this study does not consider the flow conditions in the bypass downstream of the inlet gate or the potential clogging of the bypass, e.g., by floating debris. Since the bypass is essential for successful downstream fish migration [30,39,43–45], further research is required to understand how the design influences downstream fish migration at HBR-BSs in more detail.

Third, the use of the porous medium combined with the developed UDF in the numerical model as a substitute for the geometrical representation of an HBR with a fine mesh resolution provides a simplified way to account for the effects that an HBR has on the flow field. Compared to the results of the model tests in the laboratory by Meister et al. [41], similar flow patterns occur upstream of the FGS with low velocities at the upstream end and high velocities at the downstream end, and only minor flow deflections due to the FGS. Moreover, Meister et al. [41] concluded that the bar shape and bar spacing have only a minor effect on the velocity field, in contrast to the HPP layout. Therefore, the choice of this method seems to be a favorable alternative regarding the computational costs. For further application, however, additional investigations and optimizations are recommended, including the capability to substitute other FGSs such as VBRs or louvers, or to consider the effects of vertical flow deflections on the FGS and vice versa as well. The same applies to the clogging of FGSs, which was neglected in this study, but could be considered in principle using the presented method. Additionally, it should be mentioned that the empirical equations for estimating the head loss coefficients used for the UDF were developed based on data mostly obtained in laboratory tests under ideal homogenous inflow conditions, and therefore the results for inhomogeneous inflows may differ [27].

Fourth, there are still uncertainties related to CFD. Numerical modeling of the hydraulic effects at a resolution that could predict fish behavior is difficult, especially due to the fact that the biological response of most fish species to the computed hydraulic

conditions is still poorly understood [57]. Therefore, the use of the RANS equations appears to be an appropriate choice for initial estimation. However, the application of these equations is associated with some limitations, especially related to rapidly changing flow conditions [56,98]. Thus, the results may differ compared to more advanced models. For future (and more detailed) studies, the application of detached eddy simulations (DES) or large eddy simulations (LES) might be considered.

Fifth, the predicted fish behavior in this study is based on the flow conditions as the output of the 3D numerical simulations as well as known behavioral relationships from the literature and the authors' experience from previous ethohydraulic experiments. However, the actual behavior of fish can differ significantly from the predicted one due to complex and unforeseeable behavior patterns [84]. For instance, little is known about the behavior of choosing the appropriate fish migration path [21]. Endogenous factors such as the motivation to migrate downstream or individual swimming performance may affect the expected fish behavior completely. Therefore, the prediction of potential fish behavior proves to be challenging. Nevertheless, future studies should consider more hydraulic parameters in the evaluation process to address existing knowledge gaps regarding fish response to varying flow conditions. As an example, in ethohydraulic experiments on upstream migration of Iberian barbel, Silva et al. [49] noted the importance of Reynolds shear stresses as a key parameter for the movements of this fish species.

### 4.3. Engineering Application Considerations

In the past, measures for both upstream and downstream migration of fish were considered as additional elements to be integrated after completion of the main structures of an HPP [24]. Therefore, upgrading existing HPPs is a complex process and unique to each site, especially related to ensuring safe downstream fish migration. Since the angled design of FGSs and low approach velocities require a large amount of space, implementation at existing HPPs often involves major construction efforts or is in some cases not feasible [34]. To avoid negative consequences for fish that may result from unsuitable FGSs and bypasses, an analysis of hydraulic conditions as well as correlations with the fish biological site-specific parameters, as presented in this study, can be crucial. Although numerical modeling can be quite extensive, it is significantly more cost-effective and less time-consuming than possible revisions after the actual implementation of appropriate measures that do not work as expected. Essential for the quality of the results are the input data from a geometric, hydraulic, and ecological point of view. Beyond that, however, technical and economic aspects also need to be considered for the practical implementation of HBR-BSs at HPPs.

In the design phase of HPPs, both new construction and modernization, model tests are often recommended to be able to consider all local site conditions affecting the flow field and bedload transport [65]. However, CFD also offers the possibility to investigate the flow conditions. In this regard, more detailed results obtained with CFD, such as the appearance of turbulent structures and their size and strength, can be used. Here, numerical simulations can offer distinct advantages due to simpler result extraction compared to real-life conditions at HPPs, especially regarding measurements of complex flow conditions such as *TKE* and *SVG*. Consequently, CFD can lead to a better understanding of fish behavior, particularly in combination with ethohydraulic studies under real-life conditions. Therefore, by combining state-of-the-art methods such as model tests, field experiments, and numerical simulations, the existing knowledge may be expanded, and knowledge gaps filled. This will allow the development of improved guidelines for more efficient measures for downstream migrating fish.

### 5. Conclusions

In the present study, idealized run-of-river hydropower plants (HPPs) based on existing HPPs were provided with horizontal bar rack bypass systems (HBR-BSs) according to common guidelines, 3D numerically simulated, and subsequently evaluated regarding essential hydraulic parameters relevant for downstream migrating fish. The focus here was

on the area upstream of the fish guidance structures (FGSs) and at the bypass entrance of HPPs in the block-type layout. Furthermore, geometric variations of key components were performed to examine their effect on the hydraulic parameters and associated potential fish behavior. The analysis of the results can be summarized with the following key statements:

- The block-type layout may lead to large flow deflections towards the turbines, resulting in spatially distinct approach flow conditions to FGSs. Therefore, the use of mean flow values in the design process (e.g., the mean rack normal flow velocity $\overline{v_n}$), as frequently applied in common guidelines, does not allow for an accurate assessment of actual conditions for downstream migrating fish.
- Complex flow conditions with relatively high values for the turbulent kinetic energy *TKE* and spatial velocity gradient *SVG*, which often caused avoidance responses in previous ethohydraulic experiments [12,48], can occur especially at the bypass entrance, but may be mostly negligible in the area upstream of the HBRs.
- The flow conditions at the bypass entrance are significantly affected by the bypass discharge $Q_{by}$, which should be determined based on the hydraulic parameters at the bypass entrance rather than a fixed percentage of the total river discharge $Q_0$, as well as by the geometric design of the entrance area and the bypass itself. In terms of fish guidance efficiency (FGE) along the FGS, the effects are negligible.
- Low rack angles $\alpha$ and the implementation of a bottom overlay can improve the FGE at block-type HPPs from a hydraulic point of view.

However, within this study, some simplifications and assumptions were necessary, thus further research is required to confirm and deepen the findings obtained. This relates, among other things, to the porous medium combined with the user-defined function (UDF) to account for the angle-dependent approach flow conditions as a substitute for FGSs in numerical models. Nevertheless, the presented procedure for evaluating and optimizing measures for downstream fish migration showed the importance of conducting such studies, as well as the increasing possibilities computational fluid dynamics (CFD) offers nowadays. In order to avoid negative effects on fish and undesirable revisions of non-functional measures for safe downstream migration of fish, detailed investigations considering site-specific conditions can be recommended in the design process of FGSs and bypass systems at HPPs.

**Supplementary Materials:** The following supporting information can be downloaded at: https://www.mdpi.com/article/10.3390/w15061042/s1, Dataset S1: collection of information for all HPPs studied.

**Author Contributions:** Conceptualization, H.Z., M.A. and B.Z.; methodology, H.Z. and B.Z.; software, H.Z. and W.D.; validation, H.Z. and W.D.; formal analysis, H.Z.; investigation, H.Z. and B.Z.; resources, M.A. and B.Z.; data curation, H.Z.; writing—original draft preparation, H.Z.; writing—review and editing, H.Z., W.D., M.A. and B.Z.; visualization, H.Z.; supervision, M.A. and B.Z; project administration, H.Z., M.A. and B.Z.; funding acquisition, M.A. All authors have read and agreed to the published version of the manuscript.

**Funding:** This research was funded by the Austrian Research Promotion Agency (FFG) with the Grant No. 865039 within the framework of the Energy Research Program (4. Call).

**Institutional Review Board Statement:** This manuscript was reviewed and approved by the University of Innsbruck and by the University of Natural Resources and Life Sciences, Vienna, Department of Water, Atmosphere and Environment, Institute of Hydrobiology and Aquatic Ecosystem Management, in accordance with its policy on objectivity in research.

**Data Availability Statement:** The data presented in this study are available upon request from the corresponding author.

**Acknowledgments:** The authors want to thank the Austrian Research Promotion Agency (FFG) for supporting this work. The computational results presented here have been achieved (in part) using the LEO HPC infrastructure of the University of Innsbruck.

**Conflicts of Interest:** The authors declare no conflict of interest. The funders had no role in the design of the study; in the collection, analyses, or interpretation of data; in the writing of the manuscript; or in the decision to publish the results.

## Abbreviations

| | |
|---|---|
| ASME | American Society of Mechanical Engineers |
| CFD | Computational fluid dynamics |
| DES | Detached eddy simulation |
| FGE | Fish guidance efficiency |
| FGS | Fish guidance structure |
| GCI | Grid convergence index |
| HBR | Horizontal bar rack |
| HBR-BS | Horizontal bar rack bypass system |
| HPP | Hydropower plant |
| LES | Large eddy simulation |
| L1–L5 | Locations 1 to 5 |
| MFS | Maximum face sizing |
| RANS | Reynolds-averaged Navier-Stokes |
| SIMPLE | Semi-Implicit Method for Pressure-Linked Equations |
| UDF | User-defined function |
| VBR | Vertical bar rack |
| V1–V8 | Variations 1 to 8 |
| WFD | Water framework directive |

**Notation**

| | |
|---|---|
| $A_{FGS,hyd}$ | Hydraulically active area of the FGS [m$^2$] |
| $h_{bo}$ | Bottom overlay height [m] |
| $h_0$ | Approach water level upstream of the HPP [m] |
| $l_{FGS}$ | Length of the FGS [m] |
| $Q_{by}$ | Bypass discharge [m$^3$/s] |
| $Q_{by,rel}$ | Relative bypass discharge [-] |
| $Q_d$ | Design discharge [m$^3$/s] |
| $Q_0$ | (Assumed) total river discharge [m$^3$/s] |
| $r$ | Grid refinement factor [-] |
| $SVG$ | Spatial velocity gradient [1/s or cm/s/cm] |
| $SVG_f$ | Spatial velocity gradient experienced by a fish [1/s or cm/s/cm] |
| $T$ | Water temperature [°C] |
| $t$ | Swimming duration [s] |
| $TKE$ | Turbulent kinetic energy [m$^2$/s$^2$] |
| $TL$ | Total fish length [m] |
| $u', v', w'$ | Local flow velocity fluctuations in $x$-, $y$-, and $z$-direction [m/s] |
| $u, v, w$ | Local flow velocities in $x$-, $y$-, and $z$-direction [m/s] |
| $v_a$ | Approach flow velocity to the FGS [m/s] |
| $v_a'$ | Outflow velocity downstream of the FGS [m/s] |
| $v_{by}$ | Velocity at the bypass entrance [m/s] |
| $v_f$ | Fish swimming speed [m/s] |
| $v_m$ | Velocity magnitude [m/s] |
| $v_n$ | Rack normal velocity component [m/s] |
| $v_p$ | Rack parallel velocity component [m/s] |
| $v_{pro}$ | Prolonged swimming speed [m/s] |
| $v_{sus}$ | Sustained swimming speed [m/s] |
| $v_0$ | Mean approach flow velocity [m/s] |
| $w_{by}$ | Bypass width [m] |
| $w_0$ | River width [m] |
| $x, y, z$ | Coordinates [-] |

| $\alpha$ | Horizontal rack angle [°] |
|---|---|
| $\alpha_w$ | Water volume fraction parameter [-] |
| $\Delta h$ | Water level difference [m] |
| $\Delta p$ | Pressure drop [Pa] |
| $\theta$ | Horizontal angle between the approach flow and FGS [°] |
| $\theta'$ | Horizontal angle between the outflow and FGS downstream of the FGS [°] |
| $\xi$ | Head loss coefficient [-] |

**Appendix A**

A mesh independency study based on the American Society of Mechanical Engineers (ASME) criteria [71] was performed to examine the influence of the mesh resolution on the flow field. For this purpose, three numerical simulations of the HPP on the alpine river with different mesh resolutions were conducted, where the MFS was halved from coarse to medium mesh and from medium to fine mesh, respectively (Table A1). As recommended by Celik et al. [71], the grid refinement factor $r$ was above the minimum value of 1.3 in both cases. Using the grid convergence index (GCI), the numerical uncertainty between two successive meshes was determined. The parameter studied was the mean flow velocity magnitude $\overline{v_m}$ at five different locations (L1–L5) in the domain, three 0.5 m upstream of the FGS (L1–L3), one behind the inlet gate in the bypass (L4), and one in the headrace channel downstream of the FGS (L5). The results are listed in Table A1. Since the GCI of the fine and medium meshes showed relatively minor differences, the resolution of the medium mesh was used for further numerical simulations.

**Table A1.** Mesh independency study using the mean flow velocity magnitude $\overline{v_m}$ at the locations 1–5 (L1–L5) from the results of the HPP on the alpine river with three different mesh resolutions, with MFS = maximum face sizing, $r$ = grid refinement factor, and GCI = grid convergence index.

| Mesh | MFS Outer/Inner Region [m] | Elements | $r$ [-] | L1 | L2 | GCI [%] L3 | L4 | L5 |
|---|---|---|---|---|---|---|---|---|
| Fine | 0.2/0.1 | 4,039,279 | | | | | | |
| Medium | 0.4/0.2 | 810,298 | 2.152 | 1.902 | 0.507 | 1.453 | 1.714 | 0.275 |
| Coarse | 0.8/0.4 | 250,128 | 1.864 | 16.305 | 3.599 | 11.614 | 3.436 | 1.477 |

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
