# Peer review of "Evaluation of Hydraulics and Downstream Fish Migration at Run-of-River Hydropower Plants with Horizontal Bar Rack Bypass Systems by Using CFD"

_water, doi:10.3390/w15061042_

Round 1
Reviewer 1 Report
Comments and Suggestions for Authors
The manuscript (MS) concerns by downstream fish migration in the hydropower plant comparing the available systems and suggested computational structure of the fish passage.
The MS was well written for the matter and solution with the limitations covering detailed analyses even though I am not expert on such architectural hydraulic solution.
The MS is accepted with the present form.
Author Response
Dear Reviewer,
Thank you very much for the review.
Best regards
Reviewer 2 Report
Comments and Suggestions for Authors
In terms of content, I have no comments. Interesting article. The type of flow was defined as turbulent, the values ​​of the Reynolds number. In the CFD model, please specify how the k and epsilon values ​​were determined.
Author Response
Dear Reviewer,
Thank you very much for the review.
Regarding your question, we used the realizable k-ε turbulence model with default model constants to determine the k and ε values, as described in Section 2.2.1. Turbulent flow conditions occurred particularly in areas relevant to the study (i.e., near the dividing piers, upstream or in the bypass, in the headrace channel).
Best regards
Reviewer 3 Report
Comments and Suggestions for Authors
The study is generally well designed and can attract the attention of readers in this field. However, there are some problems in this study. I think that the correction of these problems will contribute to the study.
To summarize briefly
-Although the study seems to be comprehensive in terms of Hydraulic Engineering, it also seems weak in terms of fish and fish ecology.
-Moreover, some corrections need to be made in some parts, especially in the introduction and conclusion sections.
-In addition to these, the spelling of the Latin fish names at the genus and species level should be written in italics.
-Other suggestions and corrections are in the attached file.

Author Response
Dear Reviewer,
Thank you very much for the review.
Please see the attachment for a point-by-point response to your comments.
Best regards
